# Spatio-Temporal Sentiment Mining of COVID-19 Arabic Social Media

**Tarek Elsaka [1,2,*], Imad Afyouni [1] 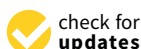, Ibrahim Hashem [1] and Zaher Al Aghbari [1]**

1    Computer Science, University of Sharjah, Sharjah 27272, United Arab Emirates
2    Agricultural Research Center, Cairo 12619, Egypt
*    Correspondence: teka@sharjah.ac.aelsa

**Abstract:** Since the recent outbreak of COVID-19, many scientists have started working on distinct challenges related to mining the available large datasets from social media as an effective asset to understand people's responses to the pandemic. This study presents a comprehensive social data mining approach to provide in-depth insights related to the COVID-19 pandemic and applied to the Arabic language. We first developed a technique to infer geospatial information from non-geotagged Arabic tweets. Secondly, a sentiment analysis mechanism at various levels of spatial granularities and separate topic scales is introduced. We applied sentiment-based classifications at various location resolutions (regions/countries) and separate topic abstraction levels (subtopics and main topics). In addition, a correlation-based analysis of Arabic tweets and the official health providers' data will be presented. Moreover, we implemented several mechanisms of topic-based analysis using occurrence-based and statistical correlation approaches. Finally, we conducted a set of experiments and visualized our results based on a combined geo-social dataset, official health records, and lockdown data worldwide. Our results show that the total percentage of location-enabled tweets has increased from 2% to 46% (about 2.5M tweets). A positive correlation between top topics (lockdown and vaccine) and the COVID-19 new cases has also been recorded, while negative feelings of Arab Twitter users were generally raised during this pandemic, on topics related to lockdown, closure, and law enforcement.

**Keywords:** Arabic tweets; COVID-19 pandemic; sentiment analysis; social data mining; spatio-temporal correlation

## 1. Introduction

Global digital statistics [1] reveal that there were more than 4.2 billion active social media users by January 2021, which is 90% of the total number of internet users. In addition, social networks have become a house for numerous real-life events that may occur in our everyday life. The global COVID-19 pandemic has been spreading worldwide, and related topics have been trending since then. Many scientists and companies have started working on challenges related to the processing and analysis of diverse types of health data, medical images, Bluetooth, and GPS data, as well as social data. From a data mining perspective, researchers have been trying to extract knowledge from people's opinions, thoughts, and feelings from social networks. Social data mining includes various associated fields such as Sentiment Analysis (SA) [2]. SA infers positive and negative mentions of people's thoughts, behaviors, and feelings based on their writings about trending topics [3]. The Twitter platform is well suited for analysing users' sentiments during the COVID-19 period, with over 353 million monthly active users [1]. Using data mining techniques, public opinion on COVID-19-related topics can be monitored and tracked in space and time.

From a different perspective, and according to the latest Internet world statistics, Arabic is ranked fourth among the ten most used languages over the Internet [4], with more than 250 million Internet users [5] originating from Arab countries. Arabic is identified by 22 Arabic-speaking countries as an official language [6]. Furthermore, millions of Arabic users

use social media networks to communicate and contribute daily Arabic content over social media. Therefore, our focus in this paper is to analyse Arabic social content available on Twitter, and to investigate people's opinions and sentiments about the COVID-19 pandemic. Recent research works have primarily focused on analyzing social data by extracting trending topics, and inferring general sentiments from related topics, with a special focus on the English language. However, COVID-19 related sentiment analysis on Arabic social media has not been fully addressed. In addition, the few existing research works on Arabic social data do not consider the spatial-temporal aspect in sentiment analysis.

In this study, we focus on analyzing Arabic social content from Twitter related to the COVID-19 pandemic, to discover people's sentiments and correlations between COVID-19-related topics and subtopics, at different levels of spatio-temporal granularities. We aim to highlight correlations between insights extracted from social data and official health data records while investigating the impact of the global pandemic on multiple aspects with different spatial and temporal scales.

This study extends our previous work [7] by presenting a comprehensive social data mining approach for the Arabic language, which employs Arabic-specific word embedding techniques with a focus on the correlation between spatio-temporal social data and official health data. Our approach presents several unique contributions compared to existing works as follows:

1. We used the dataset gathered in previous work [7] that contains Arabic tweets related to COVID-19 from two publicly shared datasets within the time frame from January 2020 to November 2020 (about 5.5M tweets). Then, we enhanced our previous approach for a location inference technique from non-geotagged tweets based on user profiles and textual content, which increased the total percentage of location-enabled tweets. We developed our Geo-Database containing bilingual (English and Arabic) names of world countries, their capitals, and the famous towns in the Arab world.
2. We implemented several mechanisms for topic-based analysis using occurrence-based and statistical correlation approaches to examine the spatio-temporal distribution of trending topics related to COVID-19.
3. We conducted a correlation-based analysis between Arabic tweets and official health data collected from online platforms.
4. We extend our previous work [7] of sentiment analysis by developing a deep learning model with bidirectional representations from the unlabeled text by conditioning on both left and right contexts in all layers. We also enhanced our detection mechanism at many spatial granularity levels (regions, countries, and cities) and different topic scales. It leverages unique insights from users' feedback on top controversial topics (lockdown, vaccination, etc.).
5. We conducted a comprehensive set of experiments and visualized our results based on the generated geo-social dataset, sentiment analysis, official health records, and lockdown data worldwide.

This paper is organized as follows: Section 2 outlines a review of some related work. The description and implementation of the proposed methodology are presented in Section 3. The results and findings of the proposed methodology are discussed in Section 4. Section 5 presents concluding remarks and future research directions.

## 2. Related Work

Current literature on social data mining has witnessed considerable achievements from NLP and ML research fields [8]. Sentiment Analysis (SA) [2] expresses the users' opinions in various forms with diverse linguistic styles to extract subjectivity and polarity from text [9] to provide countless benefits such as supporting people to make their choices. From the early days of 2020, researchers began studying social media content related to COVID-19 that focused on English tweets about COVID-19 or other Latin languages, while few researchers investigated Arabic content. Some research works motivated topic analysis to illustrate the hot topics discussed on social media using a word embedding,

word frequency, location frequency, language frequency, and character and word n-gram features weighted by TF-IDF. Meanwhile, other researchers applied feature extraction in feature-based sentiment analysis to determine sentiment polarity and forecast sentiment in social data [10]. Most of them used ML classifiers to verify results by semantic analysis. The following sections classify our review of most research studies on social streams, particularly in Arabic.

### 2.1. Data Collection and Classification

Recent research works have principally focused on analyzing social data by extracting trending topics and inferring general sentiments from related topics, with a special focus on the English language and less production on Arabic content. Many research works motivated collecting social data to be shared with the research community. In addition, they used their datasets in statistical analysis investigations such as Alanazi et al. [11] and Haouari et al. [12]. Some researchers such as Alharbi [13] identified a coronavirus dataset of Arabic tweets from three Saudi social streams and classified the dataset as conversations about precautionary steps taken by governments, conversations demonstrating social unity, and conversations endorsing government decisions. Additionally, some research works focused on analysis of tweets datasets for classification such as Hamdy et al. [14] who studied different types of tweets collected from Twitter from different perspectives of analysis and machine learning classification. They combined different machine learning models to classify tweets into related/not related to Coronavirus.

### 2.2. Geolocation Analysis

Some researchers worked on the location-enabled features of social data such as Qazi et al. [15] that introduced the GeoCoV19, a large-scale Twitter dataset related to the COVID-19 pandemic. They used the Nominatim (Open Street Maps) data at geolocation granularity levels to derive their geolocation information using a gazetteer-based method to extract toponyms from user location and tweet text. Likewise, Lamsal [16] introduced the COV19Tweets Dataset, a large-scale English language tweets dataset with sentiment ratings. They filtered the COV19Tweets Dataset's geotagged tweets to create the GeoCOV19Tweets Dataset contains only 141k tweets (0.045 percent).

### 2.3. Topic Analysis and Semantic Analysis

Alshalan et al. [17] used the ArCov-19 dataset [12], an ongoing dataset of Arabic tweets related to COVID-19, to find the hate speech in the Arab world, as well as the most common topics addressed in hate speech tweets. They used a pre-trained convolutional neural network (CNN) model to evaluate tweets for hate speech. Similarly, Alsafari et al. [18] built Arabic hate and offensive speech detection system because of an increasing proliferation of hate speech on social media. However, unfortunately, the collected data are not related to COVID-19. They applied four robust extraction algorithms based on four forms of hate: religion, race, nationality, and gender. They then labeled the corpus using a three-hierarchical annotation methodology, ensuring ground truth at each level by verifying inter-annotation agreement evaluated by applying ML classifiers. As Well, Hamoui et al. [19] examined the Arabic content on Twitter to see what the most popular topics were among Arabic users. They used Non-negative Matrix Factorization (NMF) to find the most common unigrams, bigrams, and trigrams in a dataset of Arabic tweets. They presented, discussed, and divided the final discovered topics into many categories.

Likewise, Al-Laith et al. [20] analyzed the emotional reactions of people during the COVID-19 pandemic using a rule-based technique to classify tweets. They examined six forms of emotion to discover citizens' worries. Furthermore, they created a framework for tracking people's emotions and correlating emotions with tweets mentioning some of the COVID-19 pandemic symptoms. Similarly, Bahja et al. [21] revealed the initial results of identifying the relevancy of the tweets and what Arab people tweeted about the COVID-19 feelings/emotions (Safety, Worry, and Irony). They used ML and NLP techniques to

discover what Arab people talked about COVID-19 on Twitter. Meanwhile, Essam and Abdo [22] examined how Arabs are dealing with the COVID-19 pandemic on Twitter. They extracted specific keywords and n-grams to classify common themes in the compiled corpus. They conducted a lexicon-based thematic analysis to find that tweeters had high levels of affective conversation full of negative emotions.

Some research work focused on the sentiment analysis of social data such as Manguri et al. [23]. They offered a graphical representation of the data after the sentiment analysis. Further, Chakraborty et al. [24] demonstrated tweets comprising and how health organizations have failed to guide people around this pandemic epidemic using a model with Deep Learning (DL) classifiers. Furthermore, Kabir et al. [25] created a neural network model and trained using manually labeled data to detect distinct emotions in Covid-19 tweets at fine-grained labeling. They constructed a bespoke Q&A roBERTa model to extract terms from tweets predominantly responsible for the accompanying emotions. Moreover, Hussain et al. [26] developed and used an AI-based technique to analyze social-media public reaction concerning COVID-19 vaccines in the United Kingdom and the United States to understand public opinion and discover hot subjects. They employed NLP and DL algorithms to anticipate average feelings, sentiment trends, and conversation topics. In addition, low-resource languages have witnessed recent efforts for investigating sentiment analysis, trying to bridge the gap by manually collecting and annotating social media data. ALBANA is a deep learning-based sentiment analyzer that performs sentiment analysis of around 10K Facebook comments in the Albanian language [27]. Attention mechanism along with fastText word embedding model was used to discover the interdependence and meanings of words while employing a BiLSTM for sentiment classification. Furthermore, Imran et al. [28] examined how people from various cultural backgrounds responded to COVID-19 and how they felt about the ensuing steps that various countries took in response. They used deep long short-term memory (LSTM) models to estimate the sentiment polarity and emotions from extracted tweets have been trained to reach cutting-edge accuracy. They demonstrated an original and cutting-edge method for validating the supervised DL models using Twitter tweets that had been extracted.

### 2.4. Misleading Information Detection

Some researchers tried to handle the misleading information published on social media such as Alsudias and Rayson [29] who collected and examined Arabic tweets about COVID-19 to identify the topics using the k-means algorithm, to detect rumors, and to predict tweets' sources. They used ML algorithms to identify false, correct, and irrelevant information, with two sets of features word frequency and word embedding. In a similar manner, Elhadad et al. [30] presented the COVID-19 Twitter dataset (COVID-19-FAKES) in bilingual (Arabic/English). They gathered COVID-19 pre-checked facts from several fact-checking websites to create a ground-truth database to annotate their collected dataset. They used shared knowledge from the official websites and Twitter accounts as a source of accurate information. They used ML algorithms and feature extraction techniques to annotate Tweets in the COVID-19-FAKES dataset. Similarly, Hussein et al. [31]) created an effective strategy based on the AraBERT language paradigm for combating the Tweets COVID-19 Infodemic. They trained language models on plain texts rather than tweets since pre-trained language models are widely available in many languages and available plain text corpora are larger than tweet-only corpora, allowing for greater performance.

### 2.5. Discussion

Table 1 summarizes attempts to process COVID-19-related social data with important information such as the number of tweets contained in each dataset, the language of the dataset, the time frame the data was collected, techniques used in the research work, and the features used in that work.

**Table 1.** Summary of attempts to process COVID-19 social data.

| Ref. | Purpose | Tweets | Lang. | Time Frame | Technique | Features |
|------|---------|--------|-------|-----------|-----------|----------|
| [11] | Describe Arabic tweets dataset on COVID-19 | 3,934,610 | Arabic | 1 January 2020–30 April 2020 | Statistical Analysis | Tweets frequency |
| [12] | Present and analyze Arabic Twitter dataset | 748 K | Arabic | 27 January 2020–31 March 2020 | Statistical Analysis | tweets frequency |
| [13] | Finetune a BERT model to classify multilabel tweets about crisis events | 1.6 M | Arabic | 2018–2020 | LDA model and BERT model | Topic frequency |
| [14] | Study from different perspectives of analysis, and classification | 3,934,610 | Arabic | 1 January 2020–15 April 2020 | Classification and Clustering | Word2Vec |
| [15] | Present Twitter dataset, infer geo-information and analysis dataset | 524 M | Multi-lingual | 1 February 2020–1 May 2020 | Statistical Analysis | tweet, location, and language frequency |
| [16] | Present and analyze COV19 Tweets Datasets and sentiment scores | 310 M, 141 k Geo tweets | English | 20 March 2020–17 July 2020 | Statistical and Sentiment Analysis | unigrams and bigrams, and topic modeling |
| [17] | Identify hate speech related to the COVID-19 pandemic | 547,554 | Arabic | 27 January 2020–30 August 2020 | pre-trained convolutional neural network (CNN) model | TF-IDF vectors (unigrams and bigrams) |
| [18] | Build Arabic hate and offensive speech detection system | 800,000 | Arabic and English | April to September 2019 | tweet annotation with ML classifiers | unigram, word, and char-ngrams, word embeddings and contextual word embedding |
| [19] | Examine the most popular topics raised among Arabic users | 3,934,610 | Arabic | 1 January 2020–30 April 2020 | Non-negative Matrix Factorization (NMF) | TF-IDF and Topic Coherence-Word2Vec |
| [20] | Analyze the emotional reactions of citizens | 300,000 | Arabic | 1 January 2020–30 August 2020 | rule-based technique | Emotion frequency |
| [21] | Determine the relevancy of the tweets and people feelings/emotions | 782,391 | Arabic | 16 February 2020–10 July 2020 | ML and NLP | Frequency of themes labeling |
| [22] | Analysis of Arab tweets about COVID-19 | 1,920,593 | Arabic | 1 February 2020–30 April 2020 | topics frequency and lexicon-based analysis | Most frequent features |
| [23] | Measure sentiment analysis | 500,000 | Arabic | 9 April 2020–15 April 2020 | Sentiment analysis | Word and tweet frequency |
| [24] | Demonstrate how people have tweeted regarding COVID19 | 226,668 | Arabic | 1 December 2019–31 May 2020 | Sentiment analysis | Word Vector |

**Table 1.** *Cont.*

| Ref. | Purpose | Tweets | Lang. | Time Frame | Technique | Features |
|---|---|---|---|---|---|---|
| [25] | Detect distinct emotions in COVID-19 tweets | 500 M | Arabic | 5 March 2020–31 December 2020 | neural network model | word vectors |
| [28] | detect sentiment polarity and emotion recognition | 460,286 | English | 12 February 2020–30 April 2020 | Deep long short-term memory (LSTM) | Linguistic Inquiry and Word Count |
| [29] | Identify topics, detect rumors, and predict tweets' source | 1,048,575 | Arabic | 1 December 2019–30 April 2020 | Cluster Analysis and Rumor Detection | Word frequency, count vector and TF-IDF |
| [30] | Annotate misleading Information dataset about COVID-19 Twitter dataset | 3,047,255 English and 216,209 Arabic | English and Arabic | 4 February 2020–10 March 2020 | Statistical Analysis and ML | TF, TF-IDF- (N-gram, character level) |
| [31] | Analyse social-media public sentiment in the UK and the US towards COVID-19 vaccinations | 300M | English | 1 March 2020–22 November 2020 | DL BERT | VADER and TextBlob |

To summarize, researchers used analysis of social media content to learn more about how people react to the COVID-19 pandemic. The exiting work has been conducted with English data, while Arabic data receives fewer contributions. Moreover, the existing research works on Arabic social data do not consider the spatial-temporal aspect of the COVID-19-related content. Furthermore, the correlation between official health data and social media material has not been fully studied.

## 3. Research Methods

In this paper, we propose a method to automatically detect and process social datasets containing Arabic tweets related to the COVID-19 pandemic using ML models and topic detection and tracking techniques. Figure 1 shows the workflow of our methodology, while the below subsections describe each phase in more detail. Because we focused on analyzing the spatio-temporal social data in Arabic tweets related to the COVID-19 pandemic, accordingly our methodology started with data collection of Arabic tweets. We found two publicly shared Arabic datasets for COVID-19 tweets [11] (3,314,859 tweets) and [12] (2,111,650 tweets). Unfortunately, both have a low number of geo-tweets, which is about 2% of the total tweets as discussed in Section 2. Thus, we decided to merge both datasets (about 5.5M tweets) to feed our experiments. Then, we processed the merged dataset to infer locations from non-geotagged tweets, thus generating a new dataset of location-enabled tweets which contains about 46% (about 2.5M tweets) of the original combined dataset.

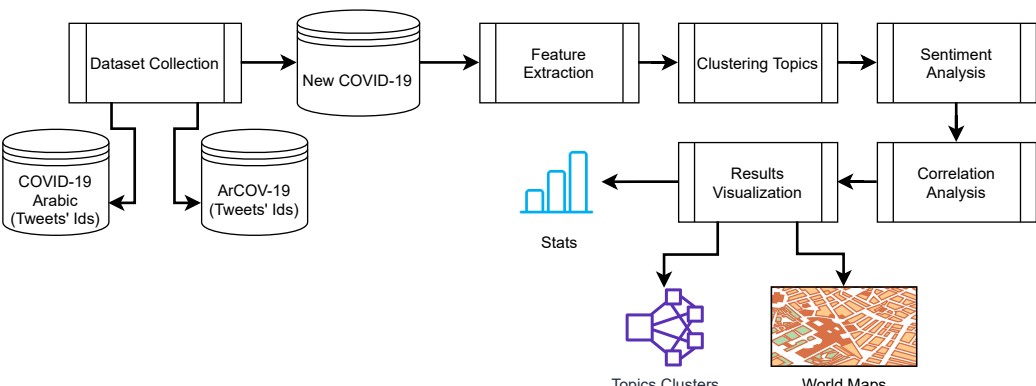

**Figure 1.** The Workflow of our methodology to analyze the social data.

Subsequently, useful features are extracted from COVID-19-related tweets. Then, classification techniques are applied to the extracted feature vectors to detect topics from the tweet's text and generate the hot topics. Then again, the sentiment analysis technique is applied with multiple perspectives to infer the sentiment and opinion polarity at many spatial (city, country, region) and temporal (day and month) levels. Next, correlation analysis between the collected tweets' data and the official health records is applied at different spatial granularities, such as country and region. Finally, we demonstrate the visual analytics based on a comprehensive set of experiments use the new Arabic COVID-19 dataset. The following subsections describe each step in our methodology.

### 3.1. Dataset Collection

The Twitter platform is well suited for studying the sentiment of users during COVID-19, with over 353 million monthly active users. Therefore we used Twitter because it focuses on becoming a broadcast platform similar to a real-time news station, with posts by default being global-readable to link people to the rest of the world. Most of the public Arabic datasets related to COVID-19 contain a low number of location-enabled tweets, such as Alanazi et al. [11] and Haouari et al. [12]. Both datasets are collected using Twitter Streaming API that matched Arabic tweets with the set of COVID-19-related keywords widely used by people, news media, and official organizations such as Coronavirus, Corona, and Pandemic.

We collected tweets from both datasets and from which we built a new geo-tagged dataset. In addition, we defined the period from 1 January 2020, to 30 November 2020, to filter tweets on the merged dataset. Unfortunately, based on the privacy policy of Twitter, both shared datasets contain only tweet IDs. Therefore, we hydrated (recollected) the dataset to get the full tweet objects from Twitter using the TWARC and Hydrator Python libraries developed for this purpose, which helps to generate JSON files for each day's tweets. The new dataset contains more than 5.5M tweets (5,054,141 tweets are unique and about 2.5M geo-tweets) with an average of Words per Tweet equal to 21. Table 2 presents the statistics of the monthly distribution of unique tweets, unique words, unique hashtags, and unique user IDs.

**Table 2.** Statistics of the monthly distribution of dataset contents.

| Month | COVID-19 Arabic | ArCOV-19 | New Dataset | Total Words | Unique Words | Unique Hashtag | User IDs |
|---|---|---|---|---|---|---|---|
| January | 208,974 | 130,002 | 338,976 | 7,179,808 | 103,823 | 5046 | 103,823 |
| February | 383,474 | 178,095 | 561,569 | 11,311,723 | 188,158 | 9207 | 188,158 |
| March | 1,479,692 | 430,235 | 1,909,927 | 41,123,366 | 615,374 | 23,820 | 615,374 |
| April | 1,307,424 | 287,754 | 1,595,178 | 32,870,009 | 516,120 | 18,777 | 516,120 |
| May | 0 | 251,670 | 251,670 | 5,575,970 | 89,974 | 11,619 | 89,974 |
| June | 0 | 212,380 | 212,380 | 4,567,400 | 81,060 | 10,679 | 81,060 |
| July | 0 | 192,754 | 192,754 | 4,063,514 | 68,352 | 9340 | 68,352 |
| August | 0 | 166,825 | 166,825 | 3,487,262 | 64,475 | 8069 | 64,475 |
| September | 0 | 113,199 | 113,199 | 2,323,648 | 47,711 | 5553 | 47,711 |
| October | 0 | 105,634 | 105,634 | 2,144,079 | 43,297 | 5050 | 43,297 |
| November | 0 | 93,958 | 93,958 | 1,915,613 | 39,354 | 4477 | 39,354 |
| Total | 3,379,564 | 2,162,506 | 5,542,070 | 116,562,392 | 1,224,684 | 48,954 | 1,224,684 |

Figure A1 illustrates the monthly distribution of tweets and hashtags during the period from February to April 2020 in which most countries around the world were in COVID-19 Lockdown. One can notice the general trend of tweeting about COVID-19 was at its peak in March and then gradually reduced to normal about normal levels. It shows that the general trends of the number of tweets and hashtags were similar.

*3.2. Features Extraction*

We developed several processes for the Feature Extraction module, as shown in Figure 2. The first process "Prepare Dataset" contains several sub-processes: "Clean Dataset", "Filter Fields", and "Prepare Arabic Text" The "Clean Dataset" process removes null values from the tweet's object. On the other hand, the "Filter Fields" process removes the unnecessary fields from the tweet's metadata. Subsequently, to prepare the Arabic text in each tweet's object for further text analysis, the process "Prepare Arabic Text" has been applied to perform the following processes. Figure A2 provides a sample of Arabic tweets with their English translation after applying all of the following:

- Convert HTML to normal text, which removes the HTML tags.
- Remove links (remove all hyperlinks of advertisements, retweets, etc.)
- Remove diacritics (remove Arabic diacritics)
- Remove punctuation (remove Arabic punctuation characters)
- Normalize and Tokenize Arabic text (We used our own Python code for tokenization)
- Remove Stop Words (remove Arabic stop words based on a list contains 750 words that published by Alrefaie [32])
- Remove empty lines (remove extra empty lines and extra white spaces).

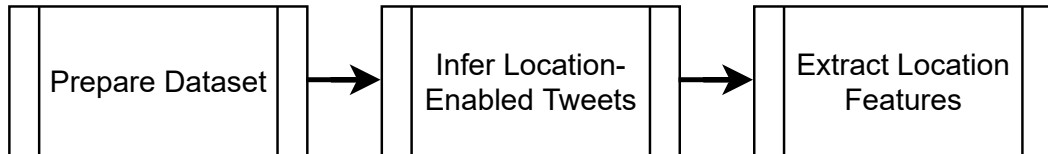

**Figure 2.** Reprocess COVID-19 tweets dataset.

Each tweet object contains several metadata fields that hold much information such as tweet text, hashtag, and user ID. The available features can be incorporated into our ML model with relative ease. The geo-location information, such as "Place" and "User Information", are important pieces of information that define the originating location of the tweet. Unfortunately, it needs further work because it is based on the user's choice to enable or disable the "Place" option in the Twitter settings. Hence, the number of geotagged tweets is so small compared to the total number of tweets. In the COVID-19 Arabic dataset [11] the number of geotagged tweets was 64,705 tweets (2%). Similarly, the ArCov-19 [12] dataset has only 50,856 tweets (2.4%). Therefore, we develop an algorithm to "generate Location-Enabled Tweets" from the non-geotagged tweets, as illustrated in Algorithm 1. The objective of the algorithm as shown in Algorithm 1 is to extract the location-enabled information. It required using the GeoDB, a manually developed Geo-location database that contains Bilingual names (English and Arabic) of world country names, capital cities, and famous towns in the Arab region. This approach analyzes all geo fields found with the tweet's object such as "Place name", "Country", and "User location" to extract the tweet source. Unfortunately, the user location information is an optional field that may is manually entered by the users. Therefore, it may be written in many languages or contained misinformation.

Our approach worked on two levels to extract chrononyms and astionyms (chrononyms: proper names of regions or countries; astionyms: proper names of towns and cities [33]) from user location. The first level tries to infer the country name from the information written in the user location metadata ([user][location]). It queries the GeoDB to extract the country name matching the ['user'] ['location']. Meanwhile, the second level works only if the first level failed, and it tries to infer the country name based on detecting the city name from the information in user location metadata and then retrieves the country name containing that city.

As a result of applying the Location Extraction algorithm, we increased the size of the experiments' dataset from about 115K (2%) up to 2.5M geotagged tweets (46%). Figure 3 illustrates the percentage of the geotagged tweets before and after applying our approach. Meanwhile, Figure A3 presents the monthly distribution of geo/non-geo tweets in our new Geo-Tweets dataset. Finally, using the occurrence-based approach, we run a process to numerically analyze the location-enabled Arabic tweets in the new geo-tweets' dataset (2.5M tweets). Most of the Arabic tweets are coming from Arab users living around the world, and most of them live in the Arab region. As a result, we compared the two regions (Arab and non-Arab) for originating the Arabic tweets that feed our experiments. This process extracted all information, which presents the proportion between Arabic tweets and top Hashtags tweeted by users in Arab and non-Arab regions. Figure 4 illustrates the comparison between Tweets and Hashtags (Top percentages only, more than 1%) in Arab and non-Arab countries during the time frame from January to November of 2020. Note that the top two countries for both tweet generation and hashtags in the Arab regions are Saudi Arabia and Kuwait. On the other hand, UK and France were the top two countries in the non-Arab region.

---

**Algorithm 1:** Location Extraction

---

**Input:**
*TD*: Tweets Dataset is the list of all tweets,
*TP*: Tweets place field, [place][name]
*TC*: Tweet Country field, [place][country]
*TCC*: Tweet Country Code field, [place][country code]
*TCO*: Tweet Coordinates field, [place][bounding box][coordinates]
*TUL*: Tweet User Location field, [user][location]
*CCDB*: Countries and Cities Database
*GeoDB*: Geo-Location Database
**Output:**
*TW*: Tweet Source

1 **begin;**
2 *TD* **= loadTweetsCorpus();**
3 **for** *each tweet in TD* **do**
4 　|　**if** *TP is not None* **then**
5 　|　　|　**country = *TC*;**
6 　|　　|　**if** *country is not None* **then**
7 　|　　|　　|　**place name = *TP*;**
8 　|　　|　　|　**country code = *TCC*;**
9 　|　　|　　|　**coordinates = *TCO*;**
10 　|　　|　**end**
11 　|　**else**
12 　|　　|　**if** *TP is None OR country is None* **then**
13 　|　　|　　|　**Select country from *CCDB* where *TUL* = country;**
14 　|　　|　　|　**if** *country is None* **then**
15 　|　　|　　|　　|　**Select country from *CCDB* where *TUL* = city;**
16 　|　　|　　|　**end**
17 　|　　|　　|　**country code, coordinates = retrieveData(GeoDB);**
18 　|　　|　**end**
19 　|　**end**
20 **end**

---

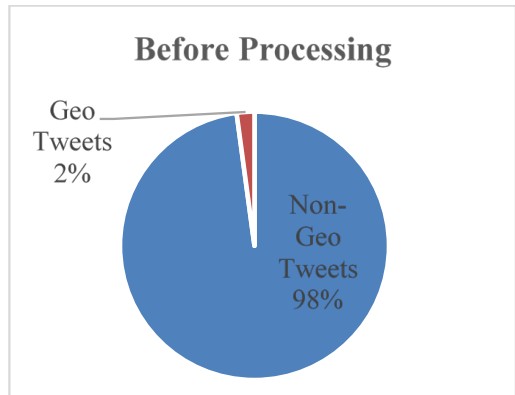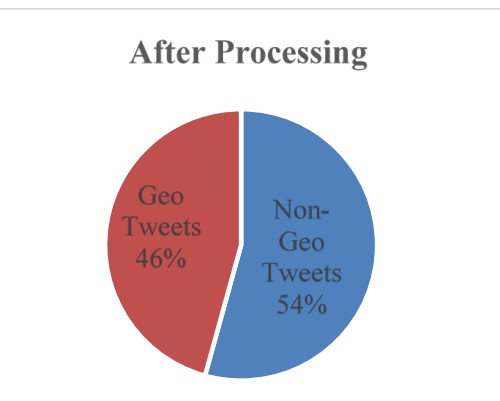

**Figure 3.** Percentage of the Geo-tagged tweets in the new COVID-19 Tweets Dataset after applying the Location Extraction algorithm.

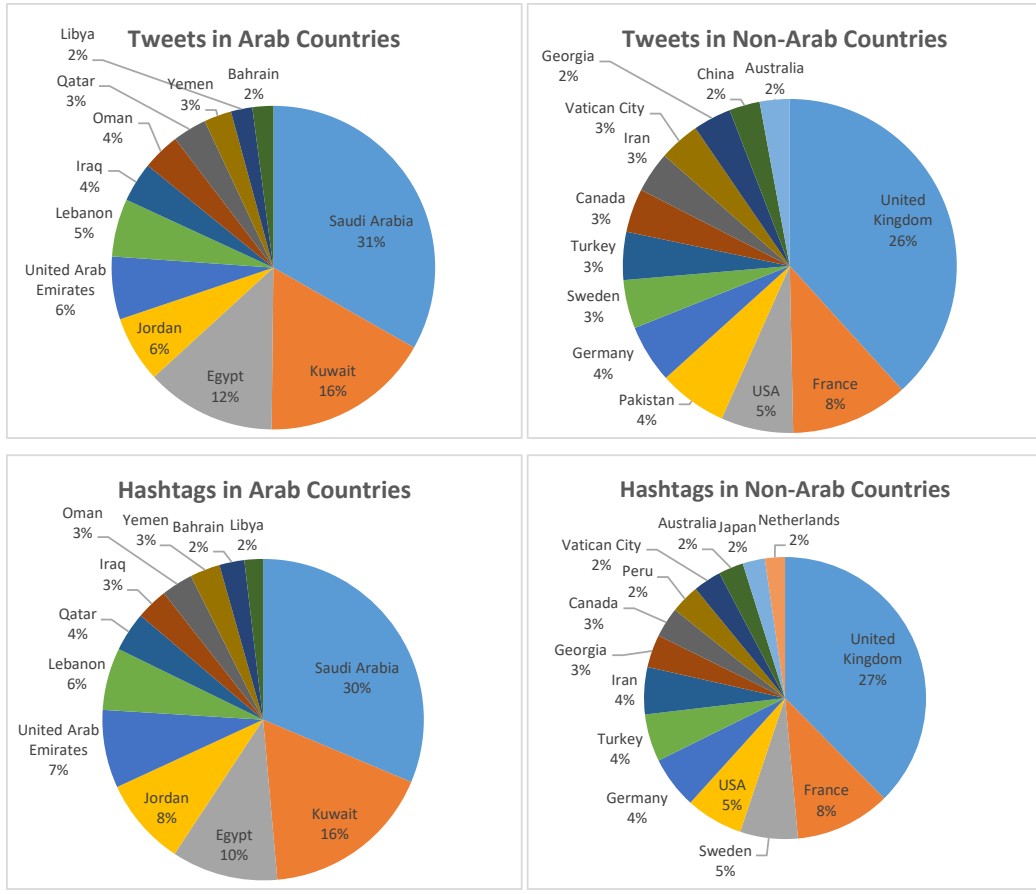

**Figure 4.** Tweets and Hashtags in Arab and non-Arab Countries.

### 3.3. Topic Clustering

In this section, we conducted some experiments to present an initial outcome of dataset distribution analysis. Our Tweets dataset was formed from two datasets originally collected using a list of keywords (about 45 unique keywords). We applied the AraVec tool to extend the list of keywords based on semantic matching. AraVec is a word embedding open-source project that aims to provide free and efficient word embedding models to the Arabic NLP research community [34]. The new list contains more than 1600 keywords. Then, we filtered the list (160 unique topics only) by removing topics non-related to COVID-19 hot topics such as names of countries and cities. Then, we applied an occurrence-based technique to each tweet text to find the best-matched keywords that represent the topics illustrated by tweets.

We used the FuzzyWuzzy Python library (an open-source string similarity matching library) to determine string similarity as a degree out of 100. It uses the Levenshtein Distance that calculates the differences between sentences. It also uses the Fuzzy C-Means (FCM) clustering method to generate a recommender system predicated on homogeneous attribute measures [35]. Figure 5 presents the monthly distribution of the top topics during the time frame from January to November 2020. Meanwhile, Figure 6 illustrates the comparison between the top 20 topics tweeted by users in Arab and non-Arab countries. Figure A4 displays the word cloud of the top topics. Figure A5 shows the comparison between top five topics in Arab and non-Arab countries. While, Figure A6 illustrates the monthly top topics distribution. The top five topics in the top 10 countries is illustrated in Figure A7. Figures A8 and A9 present the distribution of some main topics (Vaccine and Treatment) in Arab and non-Arab countries.

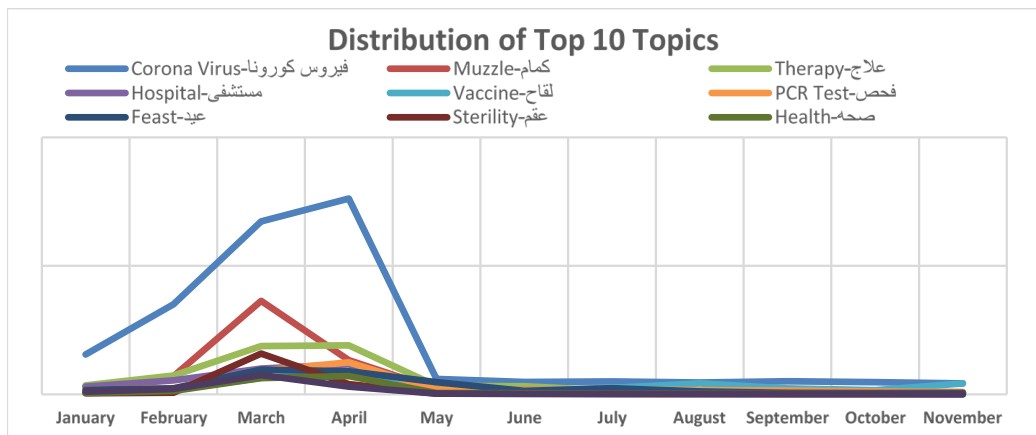

**Figure 5.** Monthly distribution of Top 10 Topics in the Tweets Dataset.

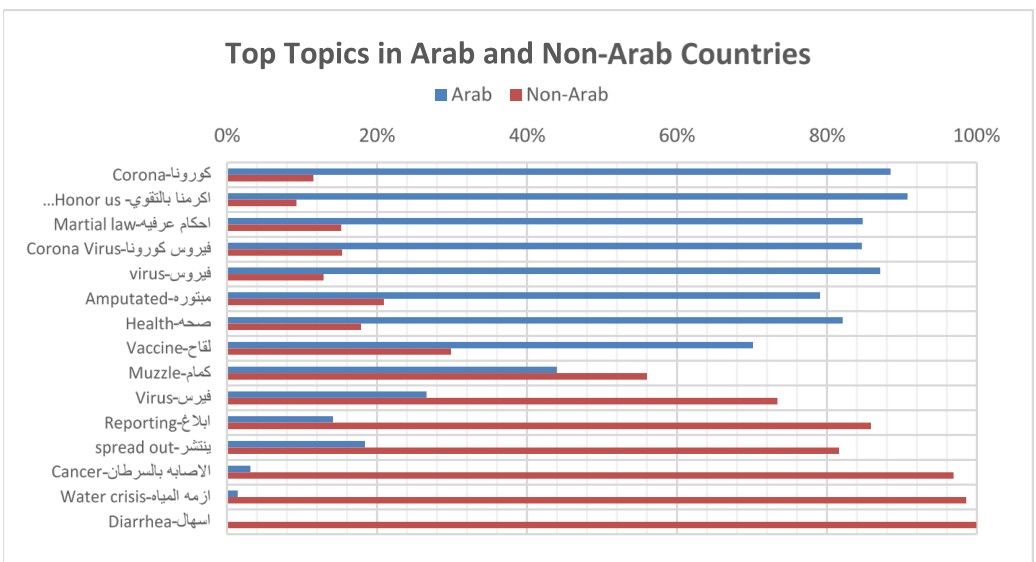

**Figure 6.** Top topics distributed globally.

Subsequently, to have a higher-level vision of topics found in Arabic tweets, we have used clustering techniques to cluster the 140 top topics extracted from our experiments into main topics. We applied the FCM clustering method that extracts 14 clusters to represent main topics. Table 3 represents the main high-level topics and their coverage. The coverage of a topic is calculated from the distribution of top topics in Arabic tweets in our Geo-dataset.

Figure 7 shows the percentage of main topics distribution in Arab and Non-countries. It illustrates that the top topics in Arab countries are Corona, Prayer, Lockdown, Hospital, and vaccines. Meanwhile, in non-Arab countries, the top topics are Sterilization, Organization, Feeling, Disease Symptoms, and Treatment. Figures A8 show Treatment topics tweeted in the Arab countries more than Vaccine topics during the first half of the year 2020 and vice versa. Meanwhile, Figure A9 displays the Vaccine topics appeared on social media in non-Arab countries from the second quarter of 2020.

**Table 3.** Main topics clusters.

| Main Topic | Coverage | Description | Example of Top Topics |
|---|---|---|---|
| Corona | 55.74% | Topics related to Coronavirus | New injuries, checkup, Virus, Corona, COVID-19 |
| Prayer | 25.68% | Topics related to prayers to God usually raised by Arab people | Save us, heal me by your ability, Protect from your torment, Your generosity, and mercy |
| Lockdown | 13.90% | Topics related to actions done by governments against the COVID-19 pandemic | Curfew, School closure, Separate service, Closing |
| Hospital | 3.41% | Topics related to tweets discuss entering hospitals and their procedures | Quarantine Hospital, Isolation Hospital, Oxygen, Anesthesia |
| Disease's Symptoms | 0.44% | Topics related to symptoms of Coronavirus | fever, cough, tiredness, headache, sore throat, diarrhea |
| Diseases | 0.23% | Topics related to other diseases rather than Coronavirus | Pneumonia, Kidney failure, Nervous breakdown, Heart failure |
| Feeling | 0.16% | Topics related to feelings about Coronavirus | kindness, laugh, discontent |
| Treatment | 0.12% | Topics related to treatment of Coronavirus | Enzymes, Tamiflu, Stem cells, Hydroxy chloroquine |
| Religious | 0.11% | Topics related to religious activities | Fasting, Fasting, Pilgrimage |
| Vaccine | 0.11% | Topics related to all talks about vaccine production or distribution | Vaccine, immunization, Serum |
| Sterilization | 0.02% | Topics related to sterilization actions | Antiseptic, Chlorine, Wash off |
| Organization | 0.01% | Topics related to organizations mentions | Health organization, Reuters |
| Miscellaneous | 0.07% | Miscellaneous topics | Rationalization, make a complaint, Electricity shut down |

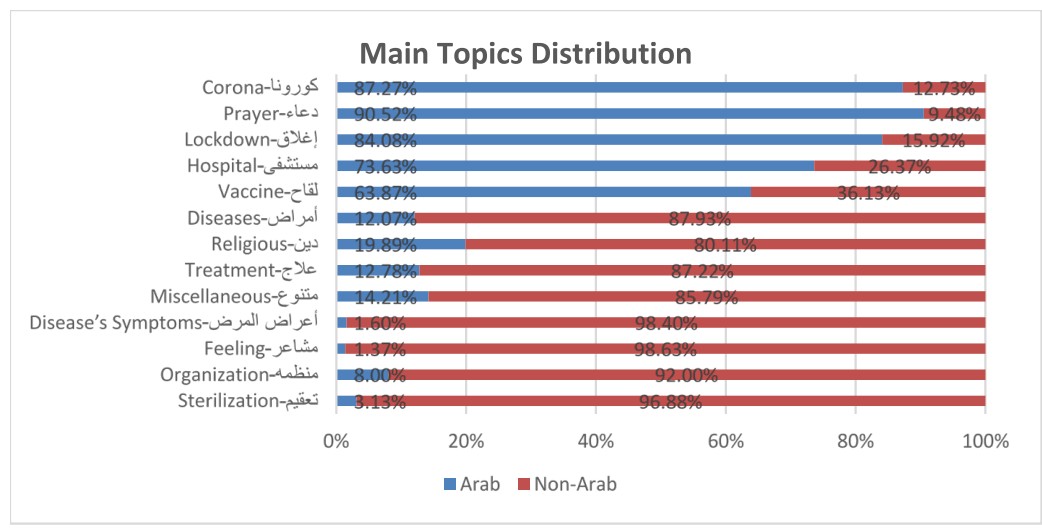

**Figure 7.** Main topics in Arab and non-Arab countries.

*3.4. Sentiment Analysis*

Social data mining has recently become an attractive field of research that relates to the method of automatically extracting valuable information from computerized textual data [36]. It includes various associated fields such as question answering systems, text summarization, and SA [2]. Generally, research on social data mining and SA can be classified into supervised and unsupervised approaches [37]. Supervised (or corpus-based) approaches require labeled datasets, usually expensive to collect and label. On the other hand, the unsupervised approach (recognized as the lexicon-based approach) depends on

extracting features for classification and clustering purposes. For instance, unsupervised sentiment analysis employs sentiment lexicons built with the assumption that words have prior sentiments. Consequently, because of the variety and wide distribution of informal writing in languages, it is a time-consuming and challenging task to create such lexicons. Many ML classifiers can be applied to improve the performance of SA on both approaches such as Decision Tree (DT), Support Vector Machine (SVM), Multinomial Naïve Bayes (NB), or deep neural networks, among others.

We used the lexicon-based approach in the SA processes with the social network data (tweets from Twitter) as an opinion resource. We built our Arabic sentiment lexicon by merging nine Arabic lexicons (Bing Liu Lexicon; NRC Emotion Lexicon; MPQA Subjectivity Lexicon; SemEval-2016 Arabic Lexicon; AEWNA Lexicon; NileULex Lexicon [8,38–42] previously tested by the research community. The Arabic sentiment lexicon contains annotation for Arabic words such as "positive", "negative", or "neutral". We used this lexicon to extract the feature of the tweet's polarity as the numbers of positive, negative, and neutral words. Then, we used polarized bag-of-words features, representing a tweet as a set of words without order, each word being a feature. Then combined these features with the number of words by polarity. Principally, the Bag of Words (BoW) technique is a word-frequency approach that counts positive, negative, and neutral words in each tweet to assign the tweet polarity as "positive" (if positive count—negative count > zero), "negative" (if positive count—negative count < zero), otherwise it is "neutral". Subsequently, we applied four ML classifiers—Decision Tree (DT), Multinomial Naïve Bayes (MNB), Linear Support Vector Machine (SVM), and Random Forest (RF)—to check the classification performance of the sentiment analysis.

To apply the sentimental analysis processes, we developed a Sentiment Extraction algorithm for the SA processes to run in our experiments. Algorithm 2 starts with loading the Arabic corpus of the text collected from Arabic tweets, which are extracted from the geo-tweets' dataset. Subsequently, the approach prepares the Arabic corpus by applying many sub-processes: (1) filtering tweets to throw away the unnecessary fields and use the necessary fields (such as: "id", "text", "country code"). (2) Remove duplicated tweets. (3) Clean tweets' text by removing special characters such as (#, &, %), hyperlinks, punctuation, numbers, Unicode characters, non-ASCII characters, and unnecessary white spaces. (4) Normalize Arabic characters (Alef, Yaa, . . . ) in tweets' text. (5) Remove Arabic stop words such as the Arabic prepositions (Men, Ela, Fe, . . . ). (6) Remove duplicated words and single-character words as well. (7) Tokenize Tweets' text. (8) Stem tweets' words to rooted words.

Subsequently, the algorithm will run on each tweet's words to get the tweet's polarity by using our Arabic sentiment lexicon to get polarity for each word then run the BoW technique to set the polarity for each tweet. Finally, the algorithm runs four ML classifiers to improve the accuracy of the SA process. The algorithm extract features and labels from the Arabic polarity corpus. Then, it divides the corpus into train and test datasets by splitting with ratios 80% and 20% consequently. Next, the algorithm vectorizes the training dataset using the TF-IDF Vectorizer. Following this, the algorithm predicts the features using the classifier model applied to the test dataset. Finally, a confusion matrix with the classifier's scores is outputted such as the precision, recall, f1-score, and accuracy values.

**Algorithm 2:** Sentiment Extraction

---

    **Input:** *TC*: Tweets Corpus,
    *AraLex*: Arabic Lexicon,
    CL: ML Classifiers' List
    **Output:** Sentiment Results

1  **begin**
2    *TC* = **loadTweetsCorpus();**
3    **RemoveDuplicatedTweets(TC);**
4    **// Prepare corpus;**
5    **for** *each tweet in TC* **do**
6        **FilterTweetsFields(tweet);**
7        **CleanTweets(tweet);**
8        **NormalizeTweets(tweet);**
9        **RemoveArabicStopWords(tweet);**
10       **RemoveduplicatedWords(tweet);**
11       **TokenizeTweetsText(tweet);**
12       **StemTweetsWords(tweet);**
13       **// Get tweet's polarity;**
14       **GetTweetPolarity(tweet, *AraLex*);**
15       **SetTweetPolarity(tweet);**
16   **end**
17  **for** *each classifier in CL* **do**
18       **ExtractFeaturesLabels(TC);**
19       **Train, Test = SplitDataset(TC);**
20       **for** *n in uni-gram* **do**
21          **Vectorising(Train, TF-IDF);**
22          **Prediction = Predict(CL, Test);**
23          **ConfusionMatrix = (precision, recall, f1-score);**
24       **end**
25  **end**

---

## 4. Results and Discussion

Figure 8 illustrates the monthly results from the sentiment analysis of COVID-19 Arabic tweets. Meanwhile, Figure A10 displays the top scores evaluating sentiment analysis in Arab countries. It shows Arab countries have the most positive/negative/neutral feelings based on users' opinions.

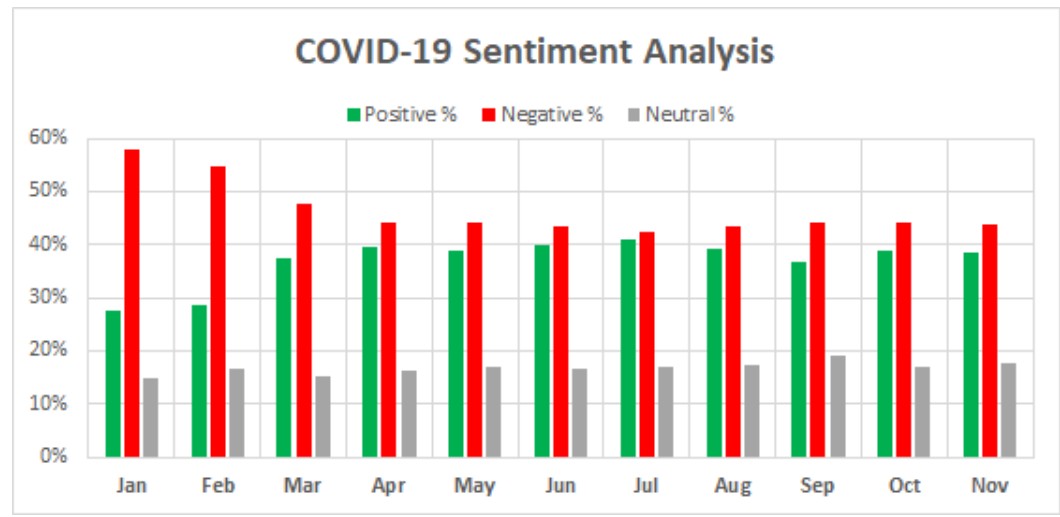

**Figure 8.** SA of COVID-19 Arabic Tweets.

Moreover, we trained four classic ML algorithms (DT, MNB, SVC, and RF) using our Geo COVID-19 dataset. Next, we evaluated the results with four sentiment-labeled Arabic datasets collected from Twitter, as shown in Table 4. Furthermore, we employed an NLP pre-training model dubbed Bidirectional Encoder Representations from Transformers (BERT), which was developed to pre-train deep bidirectional representations from the unlabeled text by conditioning on both left and right contexts in all layers [43]. We used the Arabic-BERT model (developed by Safaya et al. [44]) trained on our Geo COVID-19 dataset and evaluated with the same four datasets. We run certain model modification iterations, assess using the other datasets, and then examine how it affects the overall performance.

**Table 4.** Sentiment-labelled Arabic satasets.

| Dataset Name | Total Tweets | Pos. Tweets | Neg. Tweets |
|---|---|---|---|
| Arabic Sentiment Twitter Corpus (ASTC) [45] | 58,751 | 29,849 | 28,902 |
| Arabic Sentiment Analysis Dataset (SS2030) [46] | 4252 | 2436 | 1816 |
| 100 k Arabic Reviews [47] | 66,666 | 33,333 | 33,333 |
| Arabic Speech-Act and Sentiment Corpus of Tweets (ArSAS) [48] | 11,784 | 4400 | 7384 |

Figure 9 presents the comparison between results collected after applying ML algorithms trained and evaluated with five Arabic datasets. We used both Unigrams and Bigrams features with TF-IDF Vectorization. We found that scores are close for both, so we present unigrams only. It shows the Accuracy measure results of the traditional ML classifiers for unigram and TF-IDF feature representation. It shows that applying the SVM and RF classifiers achieved a high score on our Geo-COVID-19 dataset than using other datasets. The results achieved by some ML classifiers are also higher in performance. We applied the Arabic-BERT model and compared our findings using the five datasets that accomplished a higher accuracy than the others, especially with the BERT-mini model.

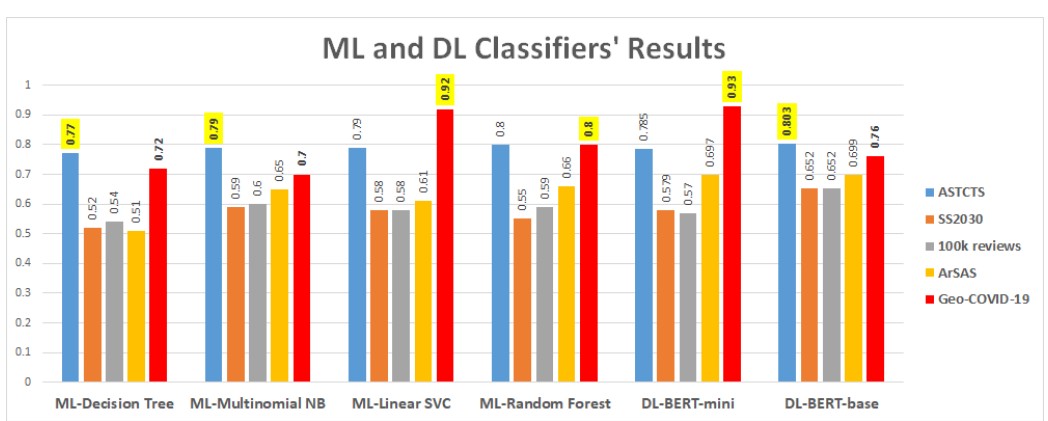

**Figure 9.** SA Classifiers' performance applied on Arabic Datasets.

### 4.1. Correlation Analysis

Daily, the official health records related to COVID-19 statistics are published by many official organizations such World Health Organization (WHO), European Centre for Disease Prevention and Control (ECDC), and Johns Hopkins University (JHU). We collected official COVID-19 records from ECDC and JHU for world countries during the time frame from January to November 2020. We implemented many mechanisms for the correlation-based analysis using occurrence-based to find the correlation between sentiment analysis, official health providers' data, lockdown information, and topic frequencies. Figure 10 illustrates the correlation between sentiment analysis of Arabic tweets related to COVID-19 and the COVID-19 new cases over the world. It shows that there were some high negative feelings at the beginning of the COVID-19 pandemic until the second quarter of the year 2020,

subsequent it was in a steady state while cases increased. In addition, Figure 11 shows separately the correlation between feelings of the Arab tweets in the Arab countries and non-Arab countries and the COVID-19 new cases with some differences in both, especially in the last quarter of 2020.

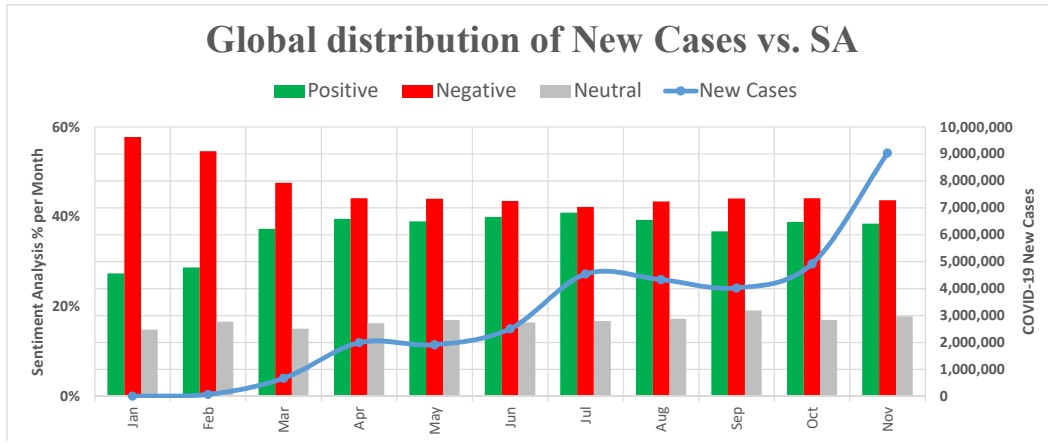

**Figure 10.** Correlation between SA and the Official Health Records.

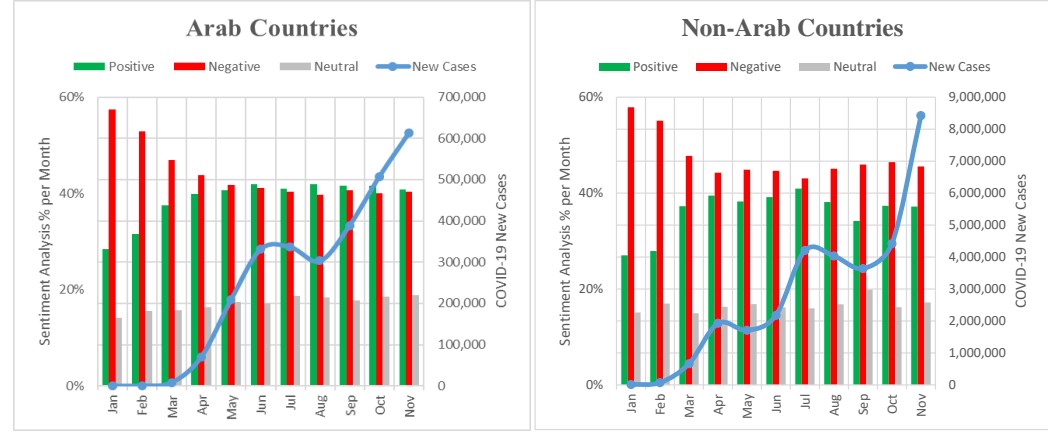

**Figure 11.** Correlation between SA and the Official Health Records in Arab and Non-Arab Countries.

Similarly, Figure A11 illustrates a considerable difference in the correlation between feelings shown in Arabic tweets and the COVID-19 new cases announced in some countries, around the world. On average most feelings are equalized between positive and negative while the number of official new cases of COVID-19 are increased. In response to the COVID-19 outbreak, governments are imposing several measures against the COVID-19 pandemic such as {School closing}, {Workplace closing}, {Cancel public events}, and {travel restrictions}. On 23 January 2020, the first COVID-19 pandemic lockdown was initiated in Wuhan [49]. Since January 2020, most governments throughout the world enacted full or partial lockdowns to stop the virus from spreading, leaving millions stranded. Moreover, a third of the world's population is restricted in some way [50]. From the middle of March 2020, several Arab countries began implementing various types of lockdowns.

Some organizations, such as Oxford University, collect data on 20 indicators to inform a Risk of Openness Index, which aims to aid countries to understand whether it is safe to "open up" or "shut down" in their fight against the coronavirus [51]. We got data from the Oxford dataset related to the lockdown status worldwide. We used data of the "School closing", "Workplace closing" and "Cancel public events" indicators as the most important indicators to feed our mechanism applied in an experiment aimed to find the correlation between lockdown and the spatial-temporal data found in our Arabic tweets' dataset. Figures 12 and 13 displays the comparison between the correlation of the lockdown with COVID-19 new cases in Arab and non-Arab countries, which present the impact of lockdown

on the spread of COVID-19 new cases. It shows a negative correlation between the number of COVID-19 new cases and the number of lockdown days in the Arab region— especially for the indicator "Cancel Public events". While, in the non-Arab region, there is a negative correlation between the number of COVID-19 new cases and the number of lockdown days—especially for the indicators "Cancel Public events" and "School Closing".

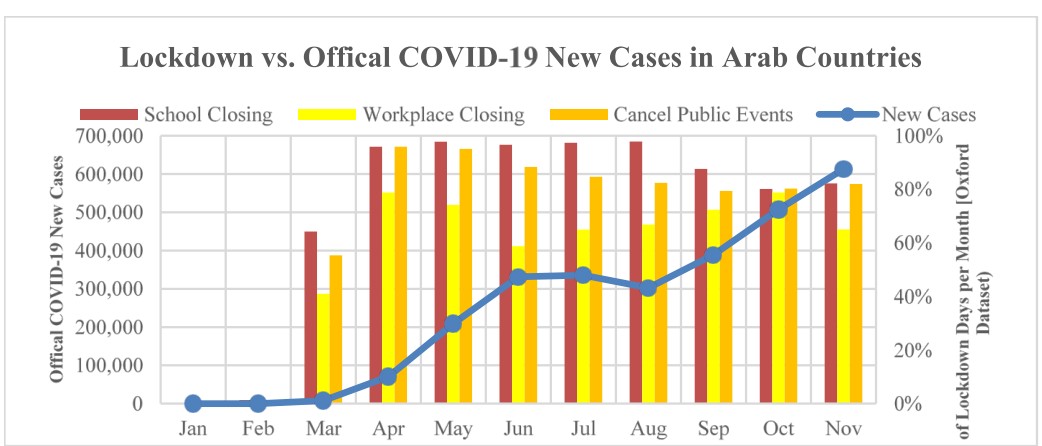

**Figure 12.** Correlation between lockdown and Official COVID-19 New Cases in Arab Countries.

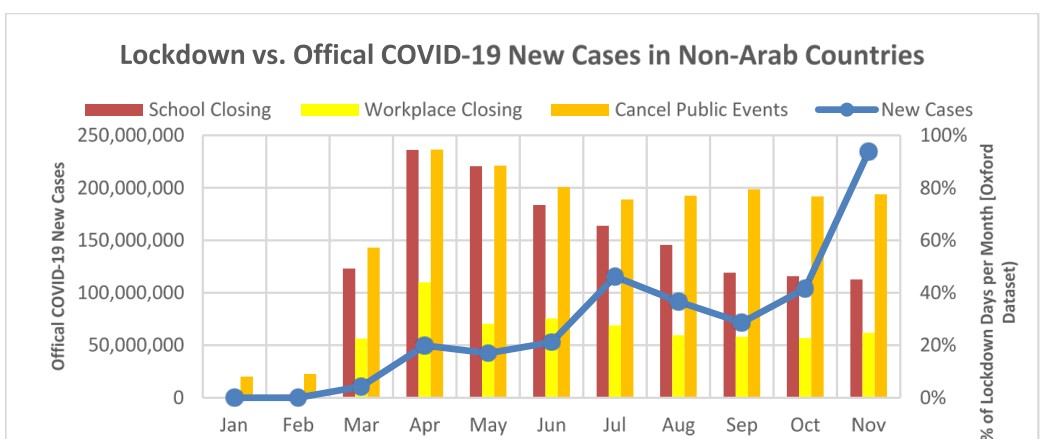

**Figure 13.** Correlation between lockdown and Official COVID-19 New Cases in Non-Arab Countries.

Figures 14 and 15 clarify the correlation between lockdown and the sentiment analysis of Arabic tweets tweeted by users during the time frame from January to November 2020, although the lockdown started late in Arab countries. The relationship shows different perspectives regarding the lockdown indicators and the sentiment of Arabic tweets related to COVID-19 in Arab and non-Arab countries. For example, they clarify a positive correlation between the positive feelings shown in Arabic tweets and the number of closing days of a school lockdown.

Back to the top-topics issue illustrated early in the above sections, we implemented another mechanism to define the correlation between sentiment analysis and topic-frequencies in a spatial-temporal manner. Subsequently, to have a wide vision of topics found in Arabic tweets, we clustered them into main top topics as discussed early in Table 3. Figures 16 and 17 illustrate the relationship between main topics and the sentiment analysis of Arabic tweets presented those topics in Arab and non-Arab countries. It shows the positive/negative feelings associated with each topic in that region.

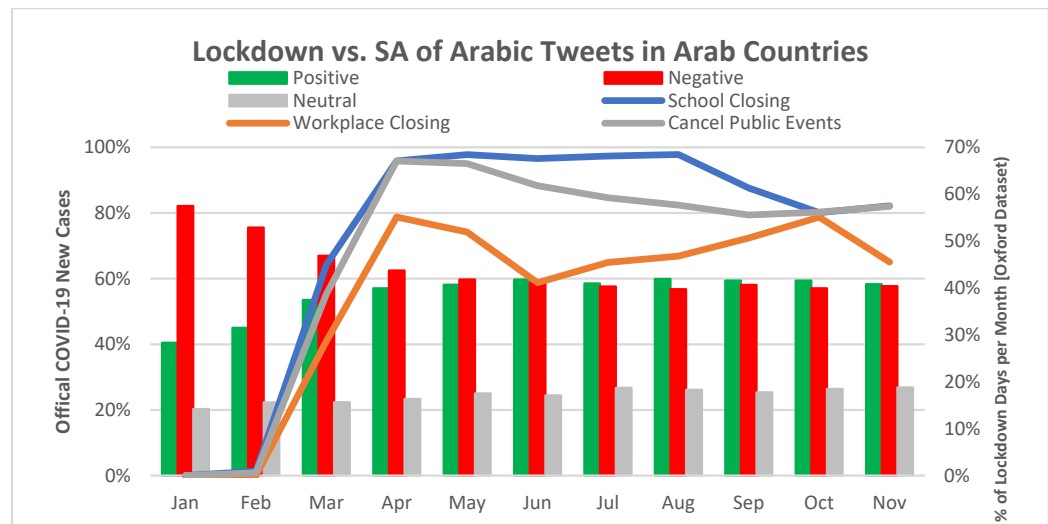

**Figure 14.** Correlation between Lockdown and SA in Arab Countries.

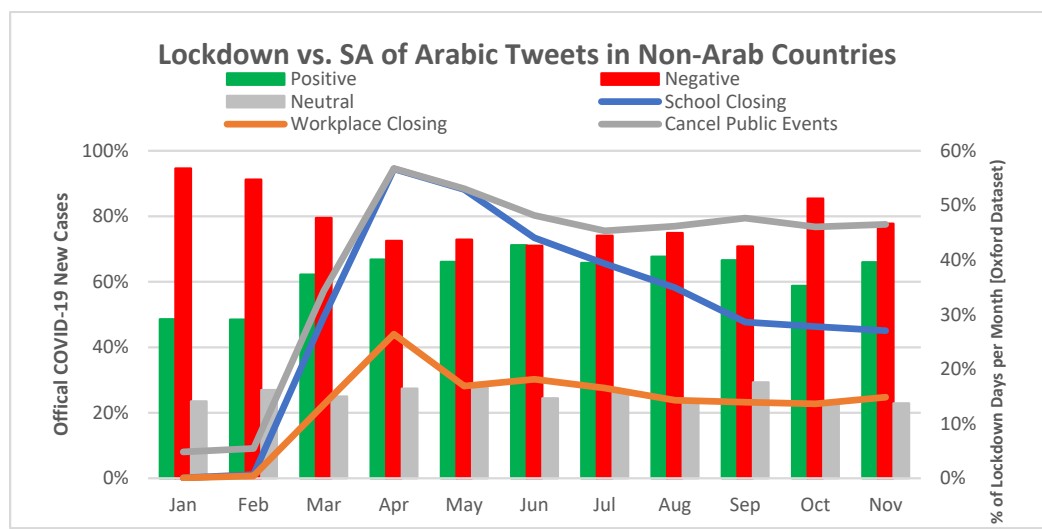

**Figure 15.** Correlation between Lockdown and SA in Non-Arab Countries.

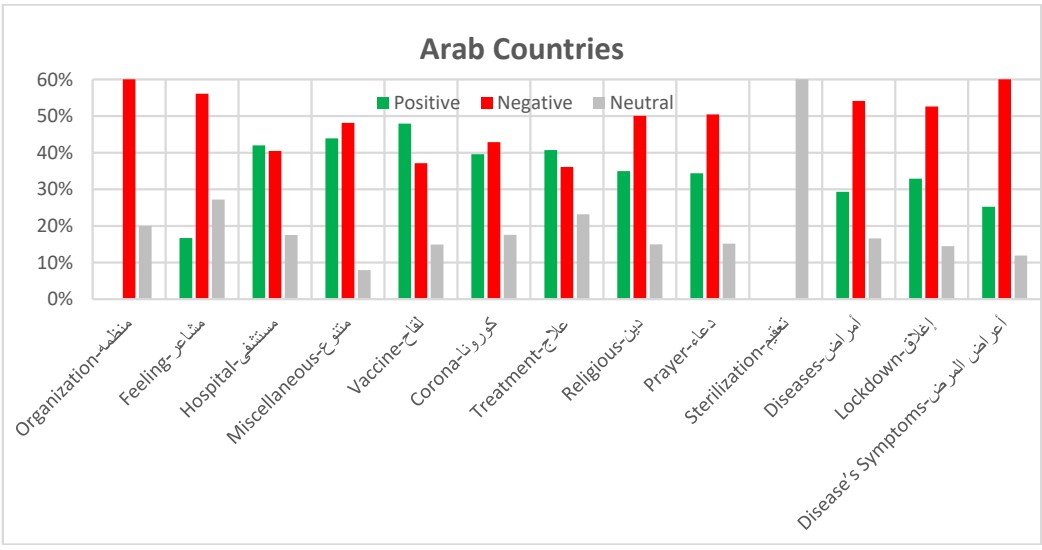

**Figure 16.** Correlation between main topics and SA in Arab countries.

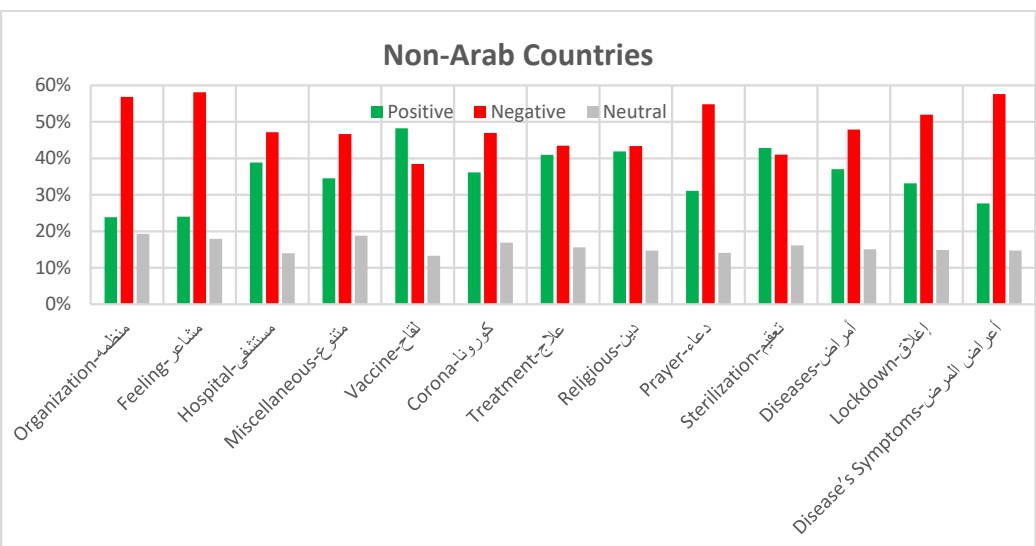

**Figure 17.** Correlation between main topics and SA in non-Arab countries.

　　We ran multiple experiments on our Arabic tweets COVID-19 dataset and the other data collected from previous experiments to explain the insight obtained from the analysis of increasingly large datasets. The results could play an essential role in representing our large-scale dataset with official data resulting early. We provided figures for visualizing data and findings in an approachable and stimulating way to enable us to extract information, superior to understand the data, and make more effective decisions. We tried to visualize the dataset components on world maps to learn COVID-19 pandemic distribution affects social media users globally. Figures A12–A14 illustrates the distribution of Arabic tweets, Arabic Hashtags, and Twitter users around the world. They clarify that most of them originated from Saudi Arabia and Egypt because they have the highest population as well the highest number of Internet users [1]. Meanwhile, Figure A15 displays the SA of Arabic tweets from our Geo-dataset over the world. It shows the average sentiment analysis of Arabic tweets is negative in most countries. Furthermore, Figure A16 illustrates the Word cloud of Sentiment Analysis over the world.

　　In addition, Figure A17 shows the distribution of COVID-19 confirmed cases globally, which emphasizes the known fact that the USA, India, and Brazil had the highest total number of COVID-19 confirmed cases. Finally, Figure A18 shows the distribution of the total number of lockdown days around the world based on the indicators used such as "Cancel Public Events" and "School closure". Specifically, it shows that most Arabic countries were in Lockdown for almost the year.

*4.2. Discussion*

　　In this research, we developed a novel location inference technique from non-geotagged tweets based on user profiles and textual content, which resulted in increasing the total percentage of geotagged tweets from 2% to 46% (about 2.5M tweets). Currently, we are conducting further research to improve the proposed location inference technique by using location inference from tweet textual content, such as the technique presented in the Arabic NER model using Flair embeddings [52]. This aims at increasing the percentage of produced geotagged tweets by utilizing an NLP process to manipulate the tweet's text. Moreover, evaluating our geo-dataset compared with four sentiment-labeled Arabic datasets (using four ML algorithms and one DL algorithm) shows that it accomplished a higher accuracy than the others, specifically with the DL model (BERT-mini).

　　In addition, some of the conducted correlation analysis shows that negative feelings of Arab Twitter users were raised during this pandemic. Moreover, we illustrated that there is a correlation between the discussed topics in the Arabic social media, such as lockdown and travel restrictions, which were enforced by governments, and the number of

COVID-19 new cases. Furthermore, the analysis showed that a positive correlation exists between the negative feeling of Arab users and the number of daily confirmed cases of COVID-19. Moreover, evaluating sentiment analysis in Arab countries shows that Saudi Arabia, Kuwait, and Egypt have the most positive/negative/neutral feelings based on users' opinions.

The correlation analysis in related work was mostly focused on time series analysis over health data, where we demonstrated in this work that spatio-temporal correlation with health and social data, can provide us with deeper insights on how people's sentiments differ over different topics and subtopics, and with different spatial scales, such as cities, countries, and regions. Finally, we visualized data and findings to understand the data components and distribute the effects of using social media related to COVID-19 globally.

## 5. Conclusions

This paper introduces a comprehensive social data mining approach for deriving COVID-19-related insights in the Arabic language, with a focus on the correlation between spatio-temporal social data and health data. In addition, it presented a sentiment analysis mechanism at multiple levels of spatial granularities and several topic scales. In addition, a technique to infer geo-information from non-geotagged tweets was developed, which increased the total percentage of location-enabled tweets from 2% to 46%, superior to most previous related works. To verify sentiment analysis performance, we applied sentiment-based classifications at many location resolutions (regions/countries) and some topic abstraction levels (subtopics and main topics) to derive people's opinions. Finally, we conducted many experiments and visualized our results based on the generated geo-social dataset, sentiment analysis, official health records, and lockdown data worldwide. Our findings show the great potential of integrating social data mining with other data sources, such as health data, to predict the evolution of such phenomena. In addition, such correlation can later be applied to other types of data such as contact tracing and GPS data, to provide an in-depth understanding of human behavior and the correlation between social and physical user interactions.

We intend to expand the dataset in the future to include more Arabic social contents to analyze the most recent periods when the social media users' focus has changed from COVID-19 in general to vaccinations. In addition, we plan to conduct further research to improve the proposed location inference technique by utilizing an NLP process to manipulate the tweet's text to increase the percentage of produced geotagged tweets.

**Author Contributions:** "Conceptualization", Imad Afyouni, Ibrahim Hashem and Zaher Al Aghbari; "Data curation", Tarek Elsaka; "Formal analysis", Tarek Elsaka and Imad Afyouni; "Investigation", Zaher Al Aghbari; "Methodology", Tarek Elsaka, Imad Afyouni and Ibrahim Hashem; "Software", Tarek Elsaka; "Supervision", Zaher Al Aghbari; "Validation", Ibrahim Hashem; "Visualization", Tarek Elsaka and Imad Afyouni; "Writing—original draft", Tarek Elsaka, Imad Afyouni, Ibrahim Hashem and Zaher Al Aghbari; "Writing—review & editing", Tarek Elsaka and Imad Afyouni All authors have read and agreed to the published version of the manuscript.

**Funding:** This research received no external funding.

**Institutional Review Board Statement:** Not applicable.

**Informed Consent Statement:** Not applicable.

**Conflicts of Interest:** The authors declare no conflict of interest.

## Abbreviations

The following abbreviations are used in this manuscript:

| | |
|---|---|
| SA | Sentiment Analysis |
| NLP | Natural Language Processing |
| ML | Machine Learning |
| CNN | Convolutional Neural Network |

| | |
|---|---|
| NMF | Matrix Factorization |
| FCM | Fuzzy C-Means |
| DT | Decision Tree |
| SVM | Support Vector Machine |
| SA | Sentiment Analysis |
| NLP | Natural Language Processing |
| ML | Machine Learning |
| CNN | Convolutional Neural Network |
| NMF | Matrix Factorization |
| FCM | Fuzzy C-Means |
| DT | Decision Tree |
| SVM | Support Vector Machine |
| MNB | Multinomial Naïve Bayes |
| BoW | Bag of Words |
| RF | Random Forest |
| WHO | World Health Organization |
| ECDC | European Centre for Disease Prevention and Control |
| JHU | Johns Hopkins University |
| ASTC | Arabic Sentiment Twitter Corpus |
| SS2030 | Arabic Sentiment Analysis Dataset |
| ArSAS | Arabic Speech-Act and Sentiment Corpus of Tweets |
| BERT | Bidirectional Encoder Representations from Transformers |

## Appendix A. More Detailed Figures

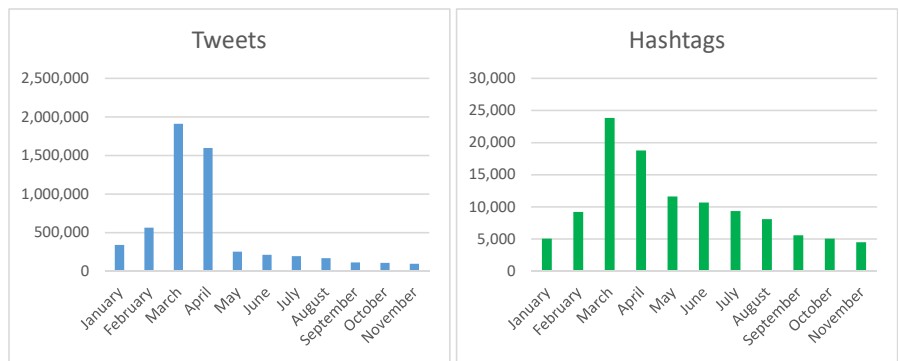

**Figure A1.** Monthly distribution of tweets and Hashtags.

| Arabic Tweet | Translated Tweet |
|---|---|
| استشارية البريقة تعلن تماثل حالتين للشفاء من كورونا | The Brega consultant announces the recovery of two cases of corona |
| عظم الله اجرك واحسن الله عزاك اخوي طارق والله يرحمه ويغفر له ويرحم جميع المسلمين انا ابوي توفي الله يرحمه ويغفر له بفايروس كورونا احترزوا ارجوكم وطبقوا التباعد الاجتماعي | May God reward you, and may God bless you, my brother Tariq, and may God have mercy on him and forgive him and have mercy on all Muslims. My father passed away. May God have mercy on him and forgive him with the Coronavirus. |
| الاحد مؤتمر صحفي للمتحدث الامني ومتحدث وزاره الصحة حول مستجدات فيروس كورونا | Sunday a press conference for the security spokesman and the Ministry of Health spokesperson on the developments of the Coronavirus |
| الزكام الكحة السخونة من اعراض كورونا ترا الشم التذوق عادي بس هل هذا يصير من اعراض | Cold, cough, fever is some of the symptoms of the corona, you see the smell, the taste is normal, but is this a symptom? |
| اليه لتصنيف الائتمان السيادي يطلقها الاتحاد الافريقي وبنك التنمية الافريقي والامم المتحدة الاليه تأتي بعد ان خفضت شركات واصبحت دوله ذات تقييم مما رفع فائده اقراضها في وقت انخفضت فيه مواردها بسبب كورونا | A mechanism for sovereign credit rating launched by the African Union, the African Development Bank and the United Nations The mechanism comes after companies reduced and became a rating country, which raised the interest of their borrowing at a time when their resources decreased due to Corona |
| الدول العظمى تفتح المدارس والدول العربية تغلق المدارس بذريعة علما بأن الاصابات في بلادهم كثيره هم دول متقدمة ونحن دول متخلفة ستستمر الدول المتقدمة بتقدمها وتستمر الدول المتخلفة بتخلفها | Great countries open schools and Arab countries close schools on the pretext that the injuries in their countries are many. They have developed countries and we are backward countries. In this way, the developed countries will continue to advance and the backward countries will continue to lag. |
| عام دراسي موفق نتمناه للجميع اليوم يشرق عام دراسي مختلف بسبب ظروف تتجدد معها الطموحات والامنيات اللهم اجعلها بداية خير لجميع الطلاب والطالبات واجعل القادم اجمل مما مضي اللهم وفقهم واكتب لهم بداية جميله وسنه دراسية سعيدة واحفظهم من كل مكروه | A successful school year we wish everyone today a different school year shines due to circumstances with which aspirations and wishes are renewed. |

**Figure A2.** Sample of Arabic tweets with English translation.

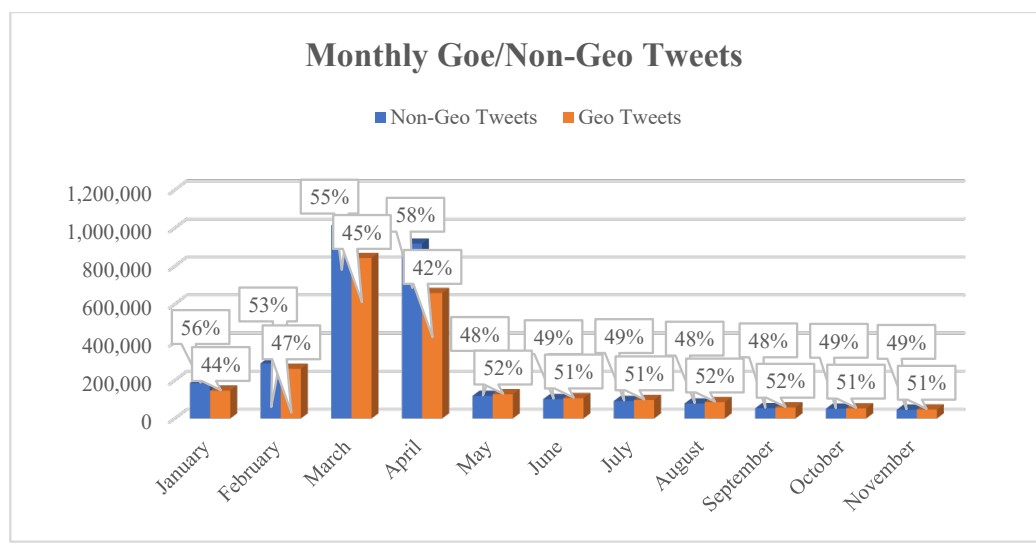

**Figure A3.** Monthly distribution of Geo and Non-geo tweets in Tweets Dataset.

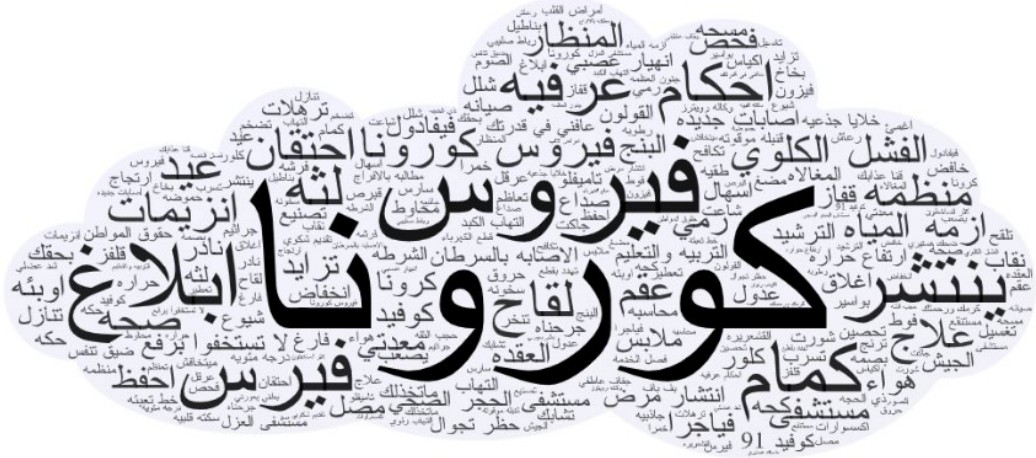

**Figure A4.** Word cloud of the top topics in the COVID-19 Arabic Tweets Dataset.

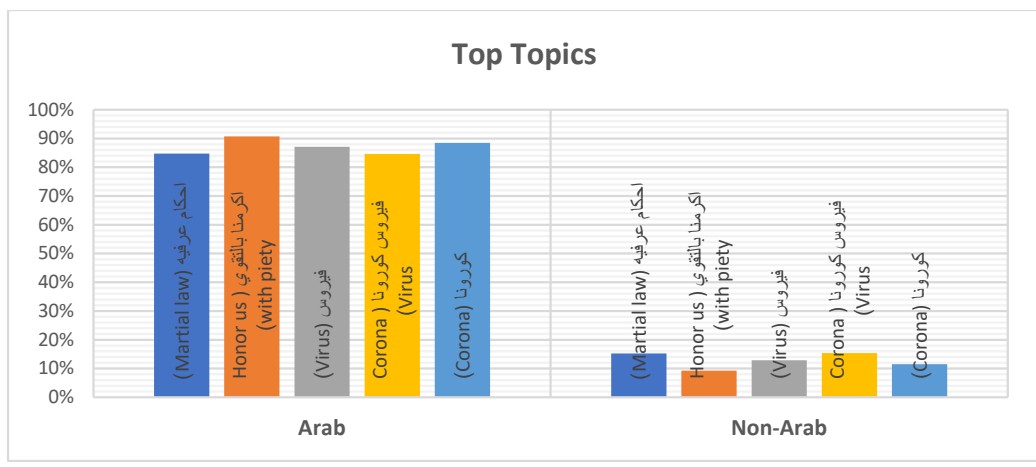

**Figure A5.** Top 5 Topics in Arab and Non-Arab Countries.

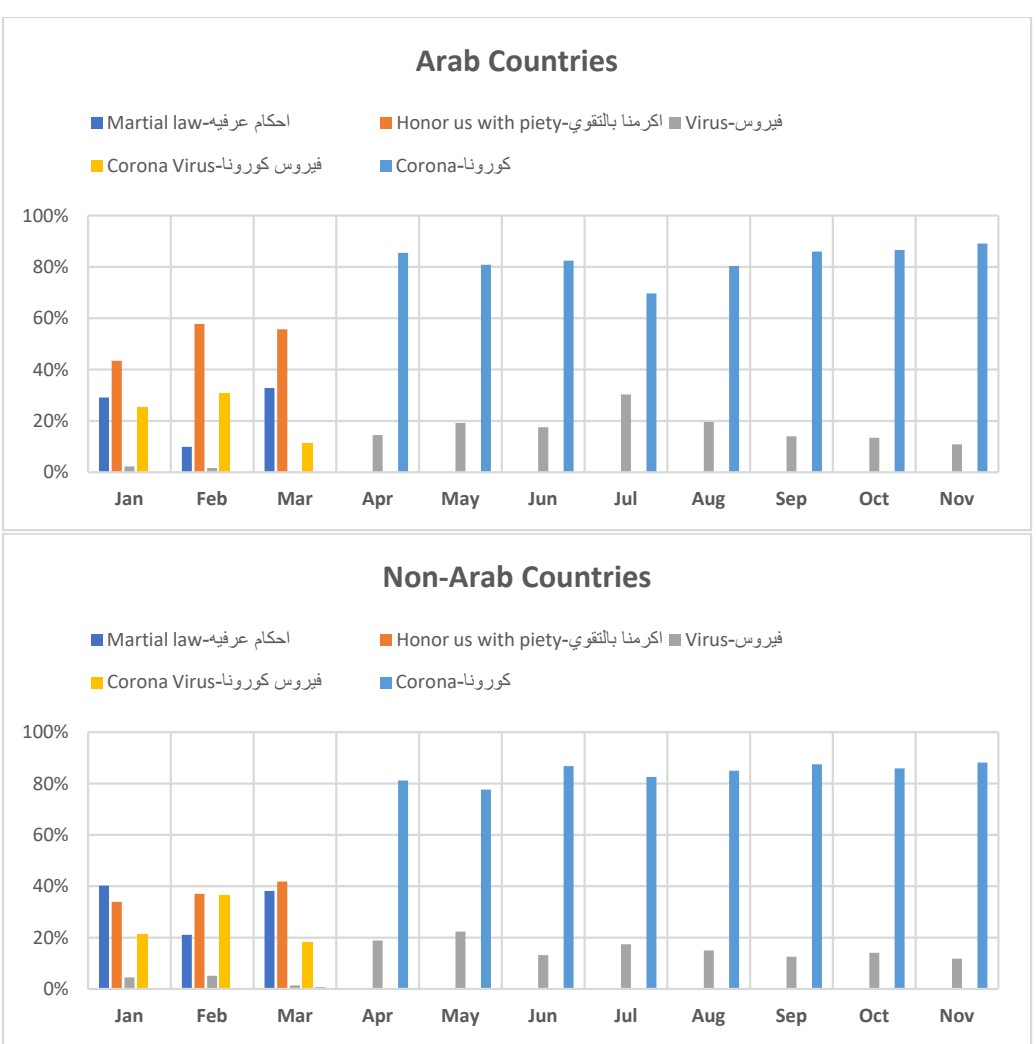

**Figure A6.** Monthly Top Topics in Arab and Non-Arab Countries.

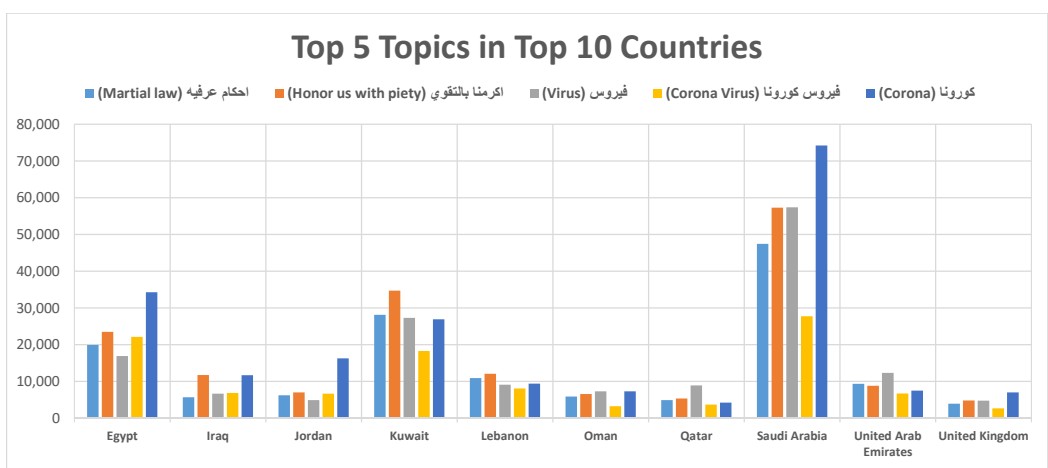

**Figure A7.** Top 5 Topics in Top 10 Countries.

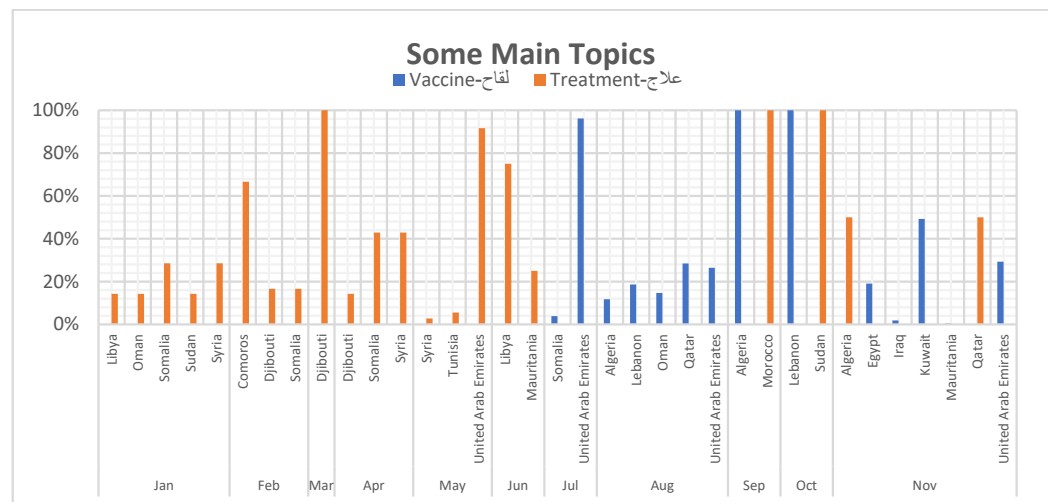

**Figure A8.** Frequency-Occurrence of Some Main Topics in Arab Countries.

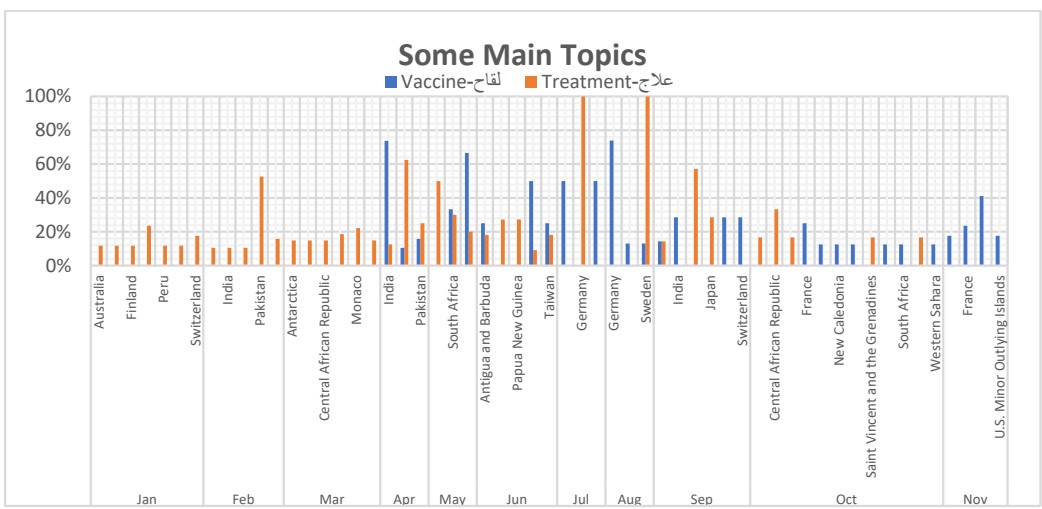

**Figure A9.** Frequency-Occurrence of Some Main Topics in Non-Arab Countries.

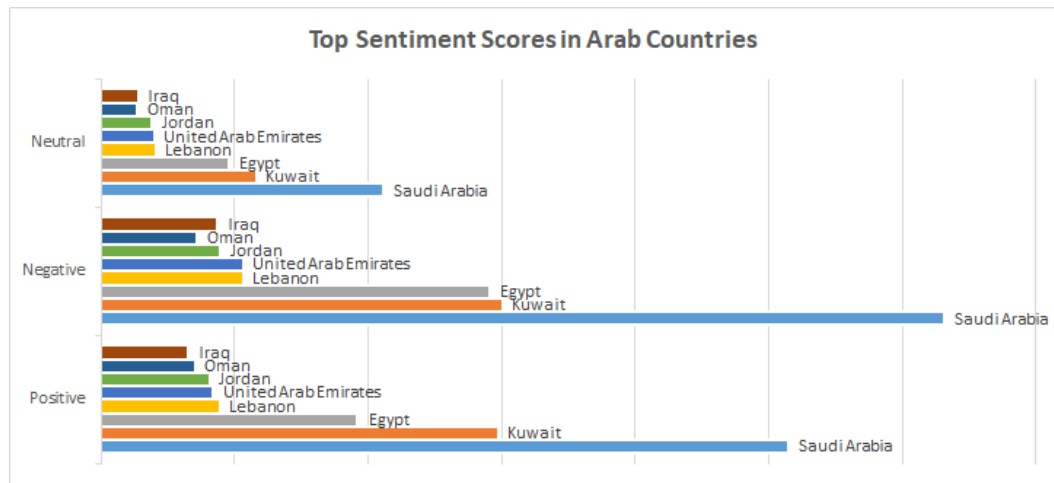

**Figure A10.** Top Sentiment (Positive/Negative/Neutral) in Arab Countries.

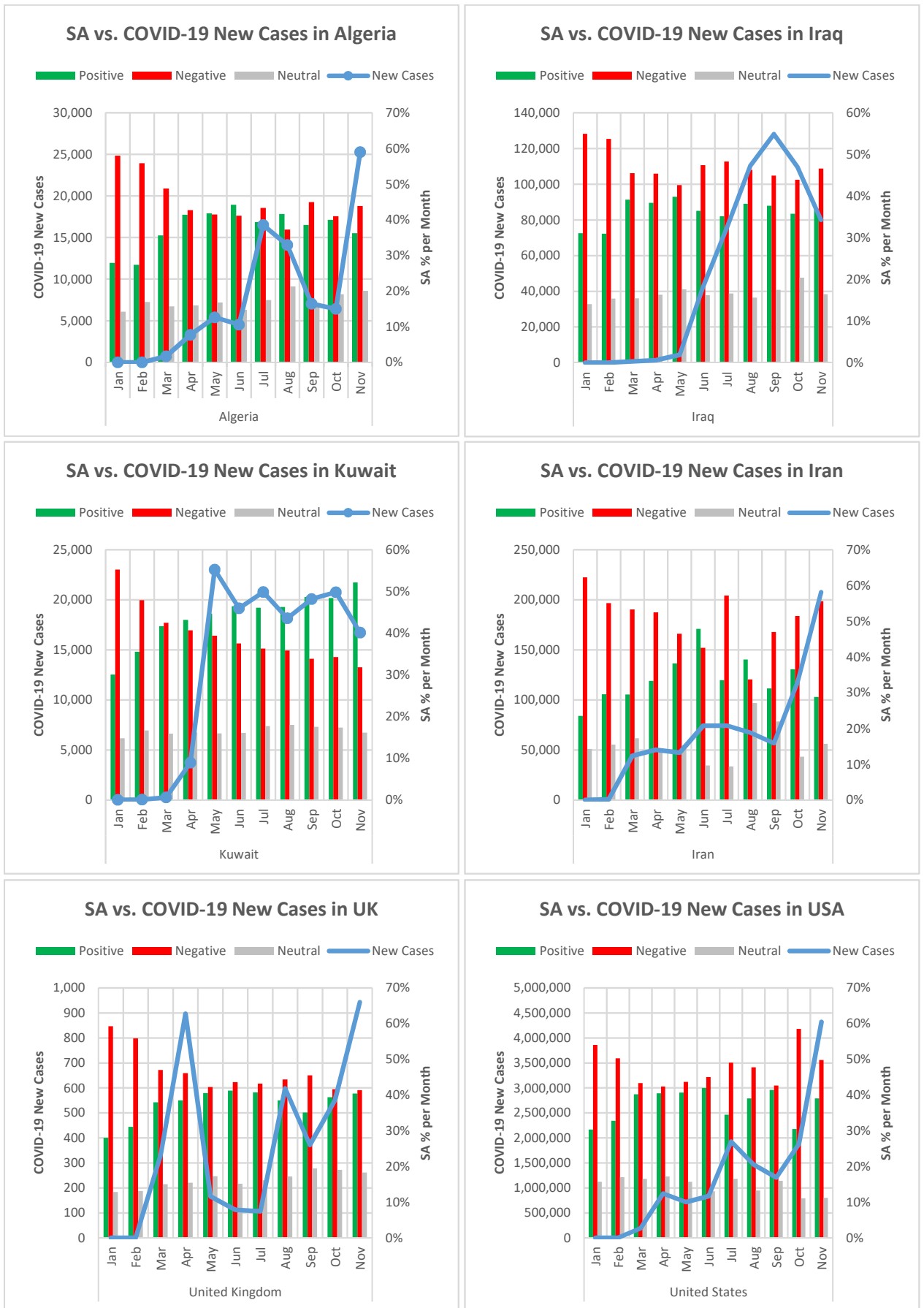

**Figure A11.** Correlation between SA and Official Health Records in some countries.

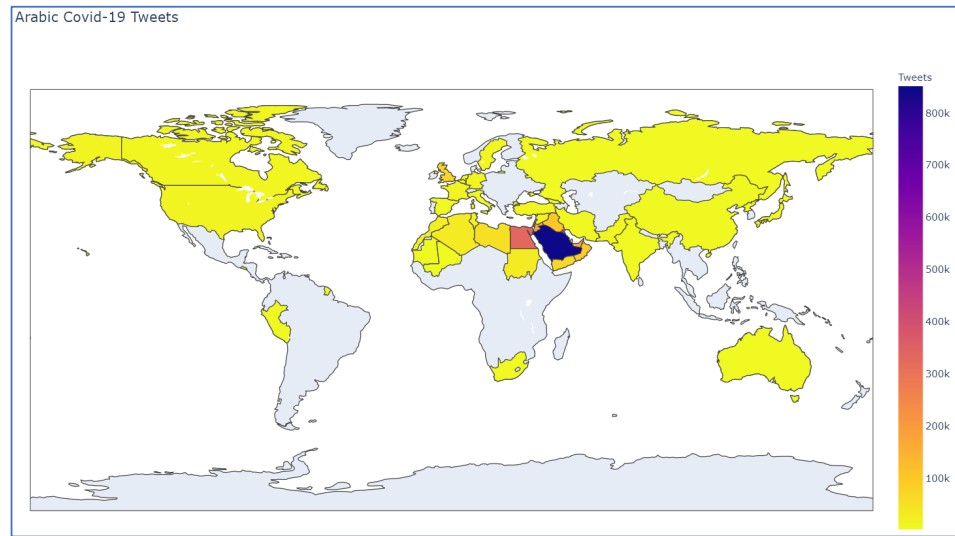

**Figure A12.** Distribution of Arabic Tweets.

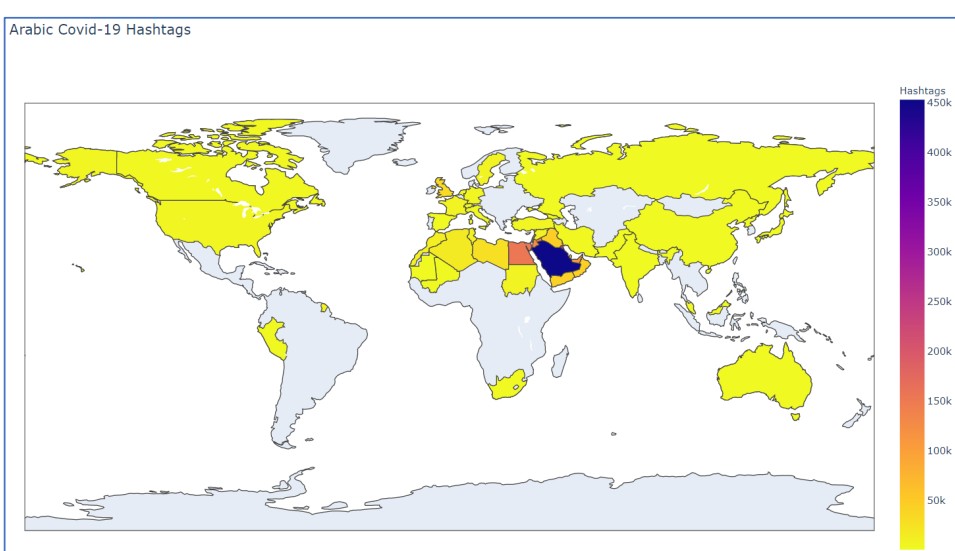

**Figure A13.** Distribution of Arabic Hashtags.

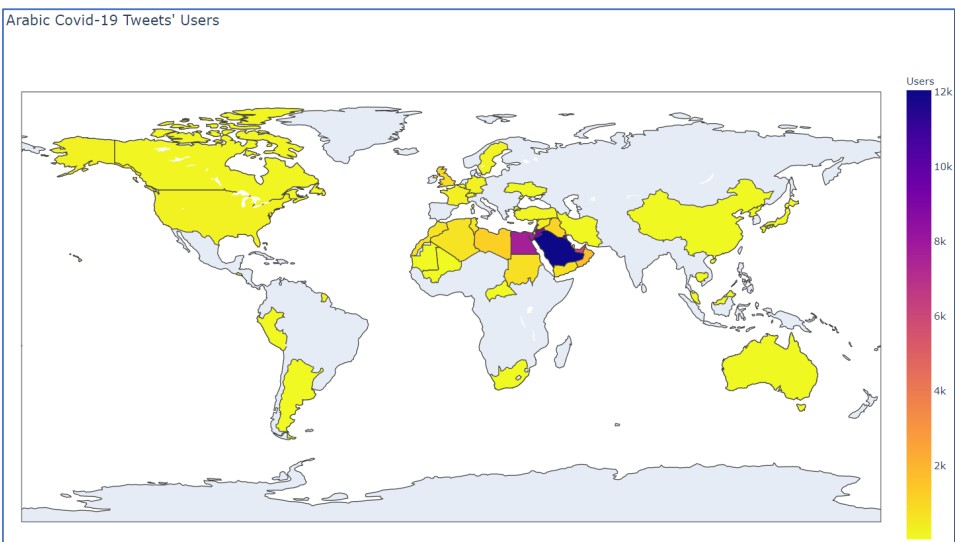

**Figure A14.** Distribution of Users.

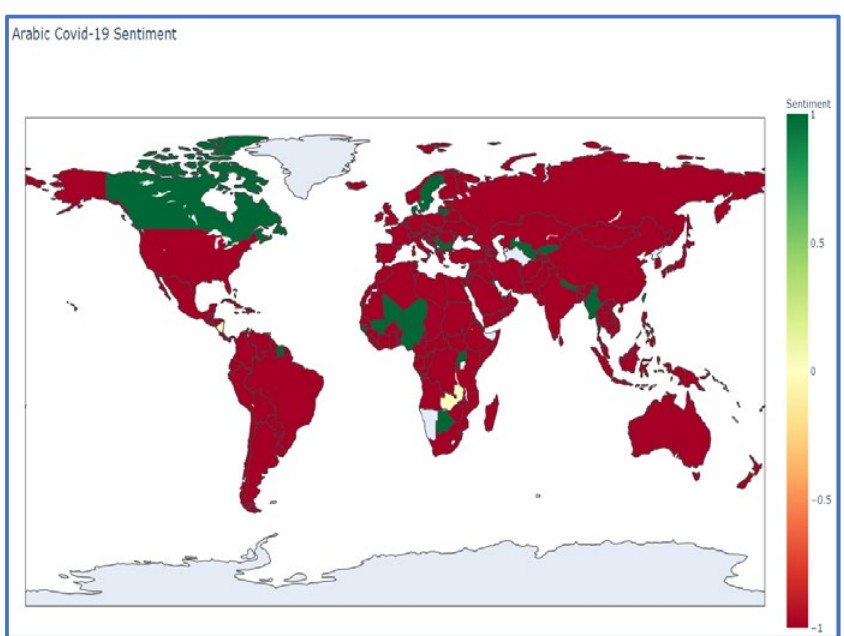

**Figure A15.** Sentiment Analysis of Arabic Tweets.

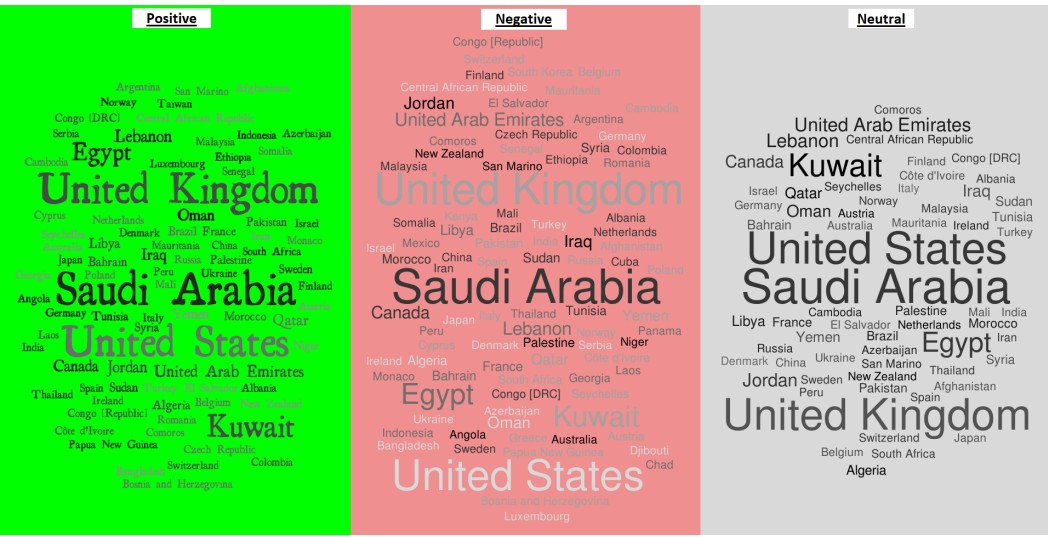

**Figure A16.** Word Cloud of Sentiment Analysis of Arabic Tweets over the world.

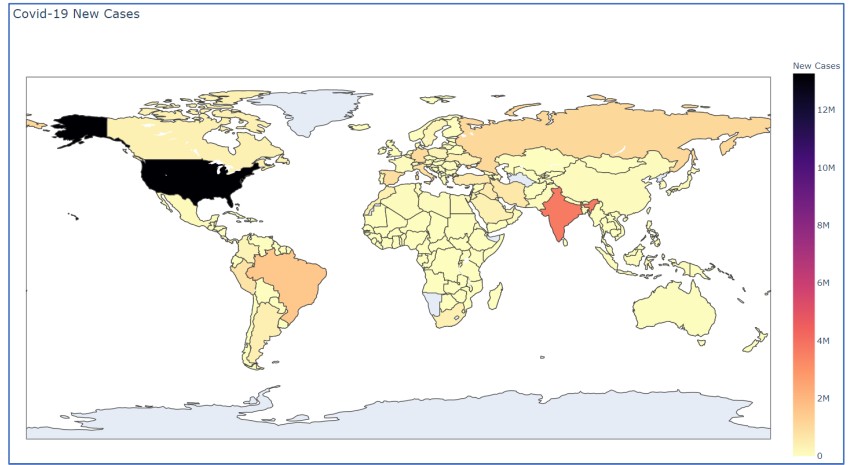

**Figure A17.** Distribution of COVID-19 Cases.

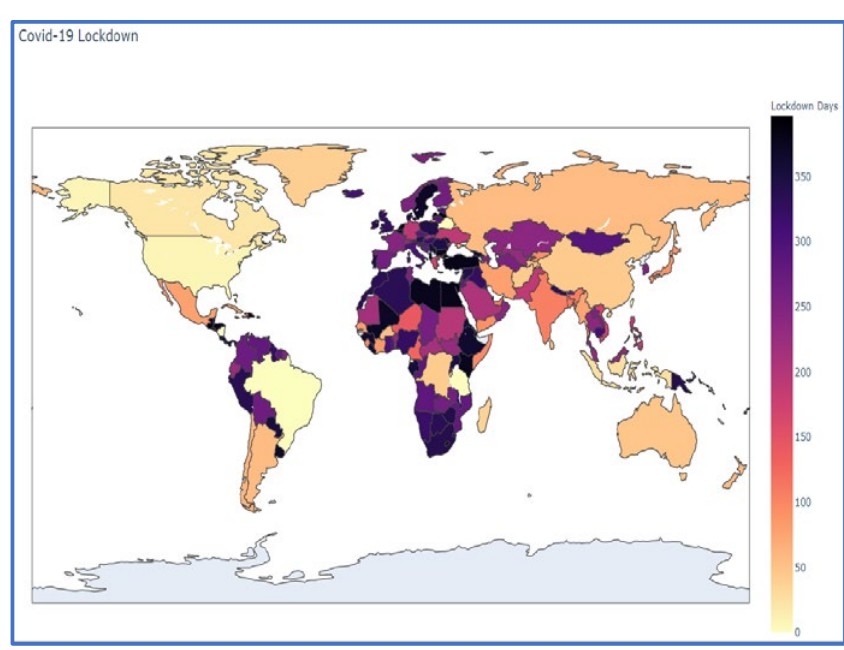

**Figure A18.** Covid-19 Lockdown Days.

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
