# Peer review of "Spatio-Temporal Sentiment Mining of COVID-19 Arabic Social Media"

_ijgi, doi:10.3390/ijgi11090476_

Round 1
Reviewer 1 Report (Previous Reviewer 2)
The comments provided in the first-round review have not been considered properly. Still introduction in not focused and unnecessary lengthy with unnecessary subheading. I also observed table in introduction which is not good.
Same comments go on methods and subsequent results and discussion section.
Discussion is too short. Compared your studies with more relevant literature.
Over all, I suggest again to follow comments provided on the first-round review.
Author Response
Response to Reviewer 1 Comments
Reviewer 1 Second Round:
Comment#1:
The comments provided in the first-round review have not been considered properly. Still introduction in not focused and unnecessary lengthy with unnecessary subheading. I also observed table in introduction which is not good.
Response: This has been thoroughly reviewed.
Comment#2:
Same comments go on methods and subsequent results and discussion section.
Response: This has been thoroughly reviewed.
Comment#3:
Discussion is too short. Compared your studies with more relevant literature.
Response: This is now adjusted, and the section is merged with the results section.
Comment#4:
Over all, I suggest again to follow comments provided on the first-round review.
Response: This has been thoroughly reviewed.
The Reviewer’s comments from the First Round:
Comment#1:
First the manuscript has not been well formatted according to MDPI ISPRS Int. J. Geo-Inf. Journal. I am not clear whether this is the research article or review article though it has been submitted as research article. I strongly suggest to have a look on the MDPI ISPRS Int. J. Geo-Inf. Journal.
Response: The manuscript has been double-checked for the formatting and latex template, and all should now be consistent with IJGI instructions.
Comment#2:
The introductory part of the manuscript is too lengthy. The authors are suggested to make introduction short and concise including research background, literature review (related works), shortcomings or research gaps and brief objectives of the manuscript. Separate heading may be not be necessary according to journal format.
Response: This has been thoroughly reviewed. The introduction is now concise and unnecessary background was removed. In addition, a clear set of contributions were added, especially to distinguish this paper from our previously published conference paper.
Comment#3:
Where is the material and methods section? Is starts from “3. Sentiment Mining of Arabic COVID-19 Tweets”. If yes clearly write material and methods. You may add other sub section under the material and methods. Same problem goes on results and discussion. Please improve it as per comment number 1.
Response: This is now adjusted, and Section 3. Is called “Research Methods” now, and it includes all necessary subsections.
Comment#4:
Again, it was very hard to understand the results section please separate it with heading Results.
Response: The results section was restructured to include all our results with a clear “Results and Discussion” heading of the section after it is merged with the Discussion section
Comment#5:
The discussion is not real discussion. I strongly suggest to compare the findings with previous research and interpret the finding accordingly.
Response: We have rewritten the discussion part to avoid any confusion, and to put our contributions in perspective with respect to existing works, especially in the location inference research field. It is now merged with the Results section.
Comment#6:
The repetition rate of this manuscript is surprisingly high. Please kindly check this in the next version of review. Are the algorithms being your own, if not please cite them properly.
Response: We have reviewed the paper for typos and duplications and have made the necessary corrections. All algorithms presented in this papers are our own, and their description is revised to make clear descriptions without repetitions.
Comment#7:
In addition to above points, the manuscript has grammatical errors and typos eg. Line 10.. “ will be presented”. Repetition of “Then” in Line 232 and 232. Please check them carefully.
Response: We reviewed the paper thoroughly for typos and grammatical mistakes.

Reviewer 2 Report (Previous Reviewer 1)
The paper is well written and can be accepted for publication
Author Response
No Comments from Reviewer #2

Reviewer 3 Report (New Reviewer)
The manuscript is centered on an interesting topic. Organization of the paper is good. A large number of experiments is conducted using the dataset collected for the purpose of the study as well as the benchmark datasets. However, there are some issues that should be addressed before the paper can get accepted for publication.
Section numbers in the last paragraph of the introduction concerning paper organization need to be fixed. (Section 5 presents discussion not conclusion)
Section 3 Methods should be named to ‘Research Method’
Discussion is rather short. It would be advisable to either merge it with the result section or elaborate it more.
The manuscript does not link well with recent literature on sentiment analysis of social media messages about covid-19 expressed in rich-resource and low-resource languages. See the recent studies ‘Cross-cultural polarity end emotion detection using sentiment analysis and deep learning on COVID-19 related tweets’ and ‘A deep learning sentiment analyser for social media comments in low-resource languages’
Author Response
Response to Reviewer 3 Comments
Comment#1:
Section numbers in the last paragraph of the introduction concerning paper organization need to be fixed. (Section 5 presents discussion not conclusion)
Response: This is now adjusted.
Comment#2:
Section 3 Methods should be named to ‘Research Method’
Response: This is now adjusted, and Section 3. Is called “Research Methods”.
Comment#3:
Discussion is rather short. It would be advisable to either merge it with the result section or elaborate it more.
Response: This is now adjusted, and the section has been extended a bit and merged with the results section.
Comment#4:
The manuscript does not link well with recent literature on sentiment analysis of social media messages about covid-19 expressed in rich-resource and low-resource languages. See the recent studies ‘Cross-cultural polarity end emotion detection using sentiment analysis and deep learning on COVID-19 related tweets’ and ‘A deep learning sentiment analyser for social media comments in low-resource languages’
Response: As suggested by the reviewer, we searched for more recent papers in the research field that are somehow relevant to the sentiment analysis of social media content. We added a new paragraph to that paper suggested by the reviewer in sections 2.3 and 2.5 besides we summarized these contributions in Table 1.

This manuscript is a resubmission of an earlier submission. The following is a list of the peer review reports and author responses from that submission.
Round 1
Reviewer 1 Report
This paper introduced a comprehensive social data mining approach for deriving COVID-19-related insights in the Arabic language, with a focus on the correlation between spatio-temporal social data and health data.
The paper should be updated with latest research studies in the field:
Hussain, A., Tahir, A., Hussain, Z., Sheikh, Z., Gogate, M., Dashtipour, K., Ali, A. and Sheikh, A., 2021. Artificial intelligence–enabled analysis of public attitudes on facebook and twitter toward covid-19 vaccines in the united kingdom and the united states: Observational study. Journal of medical Internet research, 23(4), p.e26627.
In addition, I recommend to add example of Arabic tweets along with their English translation.
Which Arab countries has most negative and positive tweets? I think further analysed required to mention which for example this Arab countries they have positive or negative comments about different aspects of COVID.
I recommend to use more recent ML algorithms such as BERT model, LSTM, CNN and BiLSTM.
What tools used for remove stop words/tokenisation in Arabic?
What other platforms such as Facebook did not used?
Can authors add the word cloud of positive, negative and neutral?
I recommend to add bigram/trigram for positive and negative respectively.
Reviewer 2 Report
The manuscript of “Spatio-temporal Sentiment Mining of COVID-19 Arabic Social Media” is an interesting research topic and has huge significance on context of ongoing COVID-19 pandemic. However, the manuscript has several limitations and could not be considered to accept for the publication. However, one chance can be given for the further improvement of the manuscript. My major concerns on the manuscript have been summarized on the following section.
- First the manuscript has not been well formatted according to MDPI ISPRS Int. J. Geo-Inf. Journal. I am not clear whether this is the research article or review article though it has been submitted as research article. I strongly suggest to have a look on the MDPI ISPRS Int. J. Geo-Inf. Journal.
- The introductory part of the manuscript is too lengthy. The authors are suggested to make introduction short and concise including research background, literature review (related works), shortcomings or research gaps and brief objectives of the manuscript. Separate heading may be not be necessary according to journal format.
- Where is the material and methods section? Is it starts from “3. Sentiment Mining of Arabic COVID-19 Tweets”. If yes clearly write material and methods. You may add other sub section under the material and methods. Same problem goes on results and discussion. Please improve it as per comment number 1.
- Again, it was very hard to understand the results section please separate it with heading Results.
- The discussion is not real discussion. I strongly suggest to compare the findings with previous research and interpret the finding accordingly.
- The repetition rate of this manuscript is surprisingly high. Please kindly check this in the next version of review. Are the algorithms being your own, if not please cite them properly.
In addition to above points, the manuscript has grammatical errors and typos eg. Line 10.. “ will be presented”. Repetition of “Then” in Line 232 and 232. Please check them carefully.
Reviewer 3 Report
The article presents an experimental analysis of Arabic social media visually, with descriptive statistics, graphs, and figures.
The biggest problem with the article is that the architecture of processing, data collection, data cleansing, supplementation with geographic coordinates, emotional analysis, and their diagrams and formal algorithms, Figure 18, can all be found in the authors’ previous work, which the authors did not mention. :
Elsaka T., Afyouni I., Hashem I.A.T., AL-Aghbari Z. (2021) Multi-scale Sentiment Analysis of Location-Enriched COVID-19 Arabic Social Data. In: Soares C., Torgo L. (eds) Discovery Science. DS 2021. Lecture Notes in Computer Science, vol 12986. Springer, Cham. https://doi.org/10.1007/978-3-030-88942-5_15
This is not acceptable in this form, it is considered self-plagiarism.
Their previous article already states that
"According to our findings, the overall percentage of location-enabled tweets has increased from 2% to 46% (about 2.5M tweets)."
In this article they write:
"Our results show that the total percentage of location-enabled tweets 13 has increased from 2% to 46% (about 2.5M tweets)."
So some of the results mentioned here are not really the result of this article, but the result of an earlier one. Previous work has been supplemented with newer analyzes, epidemiological data, a more detailed resolution, and a more detailed description of methods.
For this reason, I suggest completely rewriting the article, with an exact indication that it is an extended version of an earlier article, and to indicate exactly what the new result is compared to the previous work.
The topic of this article is interesting and current, so I definitely recommend resubmitting it after the fixes.
I would also suggest that not only visual conclusions be presented, but also more complex statistical analyzes, correlation calculations, not just simple descriptive statistics. Measurable, significant conclusions must be given.